# Using Structural Equation Modeling to Reproduce and Extend ANOVA-Based Generalizability Theory Analyses for Psychological Assessments

**Walter P. Vispoel** [1,*] **, Hyeryung Lee** [1] **, Tingting Chen** [1] **and Hyeri Hong** [2]

1 Department of Psychological and Quantitative Foundations, University of Iowa, Iowa City, IA 52242, USA; tingting-chen@uiowa.edu (T.C.)
2 Department of Curriculum and Instruction, California State University, Fresco, CA 93740, USA
* Correspondence: walter-vispoel@uiowa.edu

**Abstract:** Generalizability theory provides a comprehensive framework for determining how multiple sources of measurement error affect scores from psychological assessments and using that information to improve those assessments. Although generalizability theory designs have traditionally been analyzed using analyses of variance (ANOVA) procedures, the same analyses can be replicated and extended using structural equation models. We collected multi-occasion data from inventories measuring numerous dimensions of personality, self-concept, and socially desirable responding to compare variance components, generalizability coefficients, dependability coefficients, and proportions of universe score and measurement error variance using structural equation modeling versus ANOVA techniques. We further applied structural equation modeling techniques to continuous latent response variable metrics and derived Monte Carlo-based confidence intervals for those indices on both observed score and continuous latent response variable metrics. Results for observed scores estimated using structural equation modeling and ANOVA procedures seldom varied. Differences in reliability between raw score and continuous latent response variable metrics were much greater for scales with dichotomous responses, thereby highlighting the value of doing analyses on both metrics to evaluate gains that might be achieved by increasing response options. We provide detailed guidelines for applying the demonstrated techniques using structural equation modeling and ANOVA-based statistical software.

**Keywords:** generalizability theory; structural equation modeling; ANOVA; psychometrics; R programming; International Personality Item Pool Big Five Model Questionnaire; Self-Description Questionnaire III; Balanced Inventory of Desirable Responding; confidence intervals; continuous latent response variables

## 1. Introduction

Cronbach et al. [1] first introduced generalizability theory (GT) to the research community, and it continues to provide an elegant framework for conceptualizing how different sources of measurement error affect scores from assessment measures and how that information can be used to evaluate and improve such measures. GT techniques encompass both objectively and subjectively scored measures and can be readily applied to assessments in affective, cognitive, behavioral, and psychomotor domains. Applications of GT rely heavily on variance component estimates traditionally obtained using analysis of variance (ANOVA)-based expected mean squares within software packages catered specifically to applications of GT such as *GENOVA* [2], *urGENOVA* [3], and *EduG* [4] or from variance component programs within popular statistical packages such as SPSS, SAS, STATA, R, MATLAB, and Minitab (see, e.g., [5]). The computational framework for GT analyses also can be represented within linear mixed models (see, e.g., [6,7]). For example, in contrast to the other programs listed here, the *gtheory* package in R [8] uses the *lme4* package [9]

to fit a linear mixed model to the data. This package also uses restricted maximum likelihood (REML) rather than conventional expected mean square estimates to derive variance components, with both options also being available in most variance component programs.

Structural equation models (SEMs) offer yet another useful though less frequently applied means to estimate variance components for GT analyses in a variety of ways. Marcoulides [10] and Raykov and Marcoulides [11] were among the first to highlight such connections and demonstrate how to partially analyze one- and two-facet GT designs within SEM frameworks using LISREL [12]. Other researchers have since revisited and expanded those techniques to other designs, estimation procedures, and software packages (see, e.g., [13–24]). However, the original applications of GT to SEMs by Marcoulides and Raykov as well as those by later researchers cited here focused predominantly on derivation of variance components reflecting *relative* differences among scores for making norm-referenced decisions and typically omitted components reflecting *absolute* differences in scores for making criterion-referenced decisions. Part of the reason for such omissions was that derivation of variance components for absolute differences in scores was often considered unwieldy and fraught with technical difficulties due, for example, to presumptions that the data matrices analyzed needed to be transposed to treat facet conditions as objects of measurement and objects of measurement as facet conditions [13]. However, this method will not work in typical scenarios in which the number of persons exceeds the number of facet conditions, seemingly restricting practical uses of GT-SEMs to estimating indices of score consistency that reflect only relative differences in scores.

To overcome this perceived limitation of GT-SEMs, Jorgensen [14] proposed much simpler alternatives to obtaining variance components for absolute differences using the same GT-SEM designs analyzed in previous studies by imposing effect coding [25] and related constraints on factor loadings, means, and intercepts. However, illustrations of his procedures were based on a generated dataset of 200 normally distributed scores for a hypothetical measure of unspecified content with no clearly defined response options for items. When applying his procedures to that dataset with fully crossed one- and two-facet GT-SEM designs, using the *lavaan* SEM package in R and maximum likelihood parameter estimates, he obtained generalizability (G or $E\rho^2$) and dependability (D or $\Phi$) coefficients that varied by more than 0.003 from those produced by the anova() function in R using ANOVA mean square (MS) estimates and the *gtheory* package in R using restricted maximum likelihood (REML) estimates. After trichotomizing the original data into discrete ordered categories, Jorgensen repeated the SEM analyses using diagonally weighted least squares estimates (WLSMV in R) to place results on a continuous latent response variable (CLRV) metric that corrected indices of score consistency for possible effects of scale coarseness resulting from limited response options and/or unequal underlying intervals between those options (see [13,22]). He found that G and D coefficients were appreciably higher when taking such effects into account. Jorgensen further noted that simple commands from the *semTools* package in R [26] could be added to code for GT-SEMs within *lavaan* to produce Monte Carlo-based confidence intervals for G and D coefficients that are typically unavailable in standard GT and variance component programs. The *semTools* package also can create Monte Carlo-based confidence intervals from packages outside of R if an asymptotic sampling covariance (ACOV) matrix of variance-component parameters is available. More detailed information about Monte Carlo-based confidence intervals can be found in [27–29].

The purpose of this article is to illustrate and expand SEM procedures for analyzing fully crossed GT designs discussed by Jorgensen [14] using empirical data from respondents who completed popular self-report inventories measuring multiple dimensions of personality, self-concept, and socially desirable responding. We chose these inventories for their widespread use, strong psychometric properties, and variety of item response options. Consequently, the results provide a tangible and stronger empirical foundation for evaluating scale coarseness effects in real-life settings. The analyses also were intended to contribute new evidence of the psychometric properties of scores within GT frameworks

for the inventories administered and to extend comparisons of results between SEM- and ANOVA-based procedures beyond G and D coefficients to include variance components, proportions of individual sources of measurement error, and confidence intervals for all reported indices.

### 1.1. Background

In Cronbach et al.'s [1] original treatment of GT and in those by most subsequent authors (see, e.g., [30,31]), distinctions were made between generalizability and decision studies. In a generalizability study, researchers identify the objects of measurement and universes of admissible observations, collect data, and estimate relevant variance components. For our present applications to self-report questionnaires, persons are the objects of measurement, and items and occasions serve as possible universes of generalization. Within GT designs, universes of generalization are represented as facets that correspond to sources of measurement error that limit generalization of results. Systematic (i.e., non-error) variance in GT designs is referred to as universe or person score variance and conceptually parallels true score variance in classical test theory and communality in factor analysis (see, e.g., [20]).

In a decision study, variance components from the generalizability study are used to estimate indices of score consistency and measurement error when using scores for norm- and/or criterion-referencing purposes based on the original generalizability or altered decision study design. The most common alterations in decision studies for questionnaire data are restricting original universes of items and occasions to just items or just occasions and changing the numbers of items and/or occasions from those originally analyzed (see, e.g., [16–22,24,30,32–34]). To acquaint readers with these fundamental GT techniques for analyzing data from objectively scored self-report measures, we begin with brief introductions to relevant ANOVA-based single- and multi-facet designs and how they can be represented within SEMs.

### 1.2. Single-Facet GT Designs, Key Formulas, and Related SEMs

**Basic concepts.** Within a *persons × items* (*pi*) random effects GT design, persons and items are fully crossed, allowing the observed score for a particular person and item to be decomposed into person, item, and residual effects. The associated variance of each effect is called a *variance component*. Equations (1) and (2) show how estimated variances for item and item-mean scores are partitioned within this design.

$$pi \text{ design : Individual item score level :} \ \hat{\sigma}^2_{Y_{pi}} = \hat{\sigma}^2_p + \hat{\sigma}^2_{pi,e} + \hat{\sigma}^2_i, \tag{1}$$

$$pI \text{ design :  Item-mean score level :} \ \hat{\sigma}^2_{Y_{pI}} = \ \hat{\sigma}^2_p + \frac{\hat{\sigma}^2_{pi,e}}{n'_i}, \tag{2}$$

where $\hat{\sigma}^2$ = estimated variance component, $Y_{pi}$ = score for a particular person on a given item, $Y_{pI}$ = mean across all items for a particular person, and $n'_i$ = number of items.

Items serve as the single facet of interest here, but the same principles would apply if the facet represented other tasks, occasions, or raters. Equation (1) reveals that the overall estimated variance in scores across all items and persons is partitioned into three additive components, representing persons (or universe scores; $\hat{\sigma}^2_p$), inter-person differences in item scores plus other confounded residual error $\left(\hat{\sigma}^2_{pi,e}\right)$, and item differences ($\hat{\sigma}^2_i$). The letter I is capitalized in Equation (2) to emphasize that scores for each person are now averaged across items. The partitioning of item-mean variance across persons is more relevant in practical settings because decisions are typically made using those scores or simple transformations of them that would yield the same estimates of score consistency (e.g., multiplying item-mean scores by the number of items to obtain total scale scores). Primes appear over $n_s$ in Equation (2) and elsewhere to indicate that any number of conditions/replicates for a facet

can be specified in a decision study. The variance component for items $(\hat{\sigma}_i^2)$ drops out of Equation (2) because the mean score for items across persons in the partitioning shown is now a constant.

**Indices of score consistency and agreement.** Once estimated, the three variance components on the right side of Equation (1) can be inserted into Equations (3)–(5) to derive three key indices: G coefficients, global D coefficients, and cut-score specific D coefficients (see, e.g., [20,30,35]).

$$\hat{G} \text{ coefficient for } pI \text{ design } = \frac{\hat{\sigma}_p^2}{\hat{\sigma}_p^2 + \left(\frac{\hat{\sigma}_{pi,e}^2}{n_i'}\right)} \tag{3}$$

$$\text{Global } \hat{D} \text{ coefficient for } pI \text{ design } = \frac{\hat{\sigma}_p^2}{\hat{\sigma}_p^2 + \left(\frac{\hat{\sigma}_{pi,e}^2}{n_i'} + \frac{\hat{\sigma}_i^2}{n_i'}\right)} \tag{4}$$

$$\text{Cut-score specific } \hat{D} \text{ coefficient for } pI \text{ design} = \frac{\hat{\sigma}_p^2 + \left[\left(\overline{Y} - \text{Cut-score}\right)^2 - \hat{\sigma}_{\overline{Y}}^2\right]}{\hat{\sigma}_p^2 + \left[\left(\overline{Y} - \text{Cut-score}\right)^2 - \hat{\sigma}_{\overline{Y}}^2\right] + \left(\frac{\hat{\sigma}_{pi,e}^2}{n_i'} + \frac{\hat{\sigma}_i^2}{n_i'}\right)} \tag{5}$$

where $\hat{\sigma}_{\overline{Y}}^2 = \frac{\hat{\sigma}_p^2}{n_p'} + \frac{\hat{\sigma}_{pi,e}^2}{n_p' n_i'} + \frac{\hat{\sigma}_i^2}{n_i'}$ and corrects for bias (see [35]).

G coefficients reflect relative differences in scores used for norm-referencing purposes (e.g., rank ordering). Within the present *pI* design, they are equivalent to alpha reliability estimates [36] and would be analogous to stability or inter-rater reliability coefficients had occasions or raters been the lone facets in the design. Global and cut-score specific D coefficients take both relative and absolute differences in scores into account. Terms within parentheses in the denominators of Equations (3) and (4) represent *relative error* and *absolute error*, respectively. When item means are equal (i.e., $\hat{\sigma}_i^2 = 0$), relative and absolute error will coincide, as will G and global D coefficients. When observed scores are used for screening, selection, classification, or domain-referencing purposes, cut-score specific D coefficients provide the best indices of dependability because they reflect agreement in decisions over random repetitions of the assessment procedure [37]. Values for these coefficients will vary with the cut point chosen and increase as cut-scores deviate from the scale mean.

**SEM representation.** An SEM for the *pi* GT design based on administration of three items is shown at the top of Figure 1. This model has a single factor for person linked to each item, with factor loadings set equal to one and uniquenesses set equal. Consequently, only two variance components are directly estimated: the variance for the person factor $\left(\hat{\sigma}_p^2\right)$ and the common uniqueness across items $\left(\hat{\sigma}_{pi,e}^2\right)$. To derive the missing variance component for items $\left(\hat{\sigma}_i^2\right)$ needed to calculate D coefficients, Jorgensen [14] imposed effect coding constraints on loadings and intercepts [25] that placed results on the same scale as the original indicators (item scores here) and set the mean for the person factor equal to the grand mean of observed scores. With effect coding, item intercepts are constrained to sum to zero and factor loadings are constrained to average one (or equivalently sum to equal the number of items). Under these conditions within the present model, Jorgensen noted that $\hat{\sigma}_i^2$ can be derived using Equation (6).

$$\hat{\sigma}_i^2 = \frac{1}{n_i - 1} \sum_1^{n_i} (Intercept_i)^2 \tag{6}$$

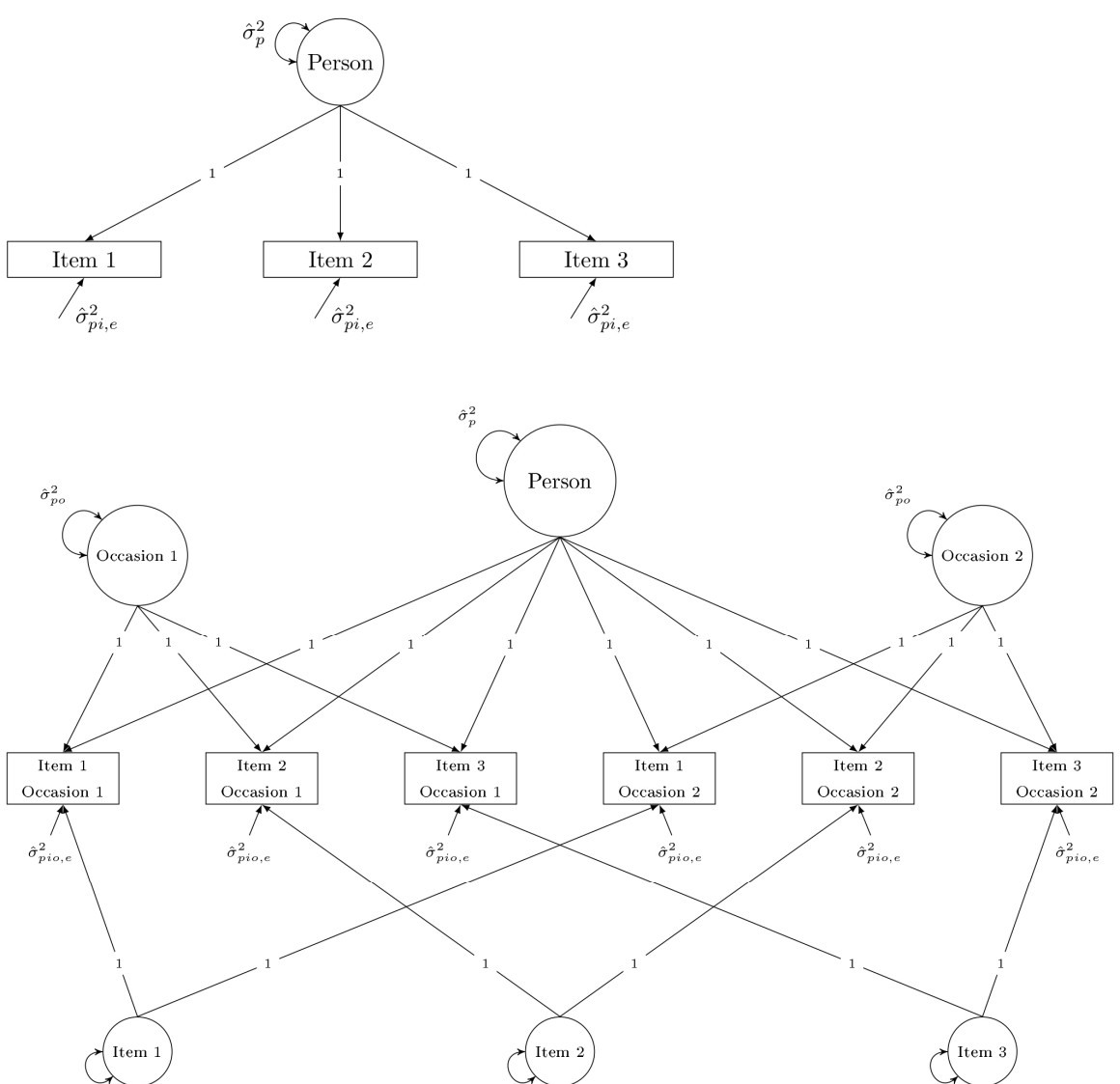

**Figure 1.** GT *pi* Design SEM with Three Items (**top**); GT *pio* Design SEM with Three Items and Two Occasions (**bottom**).

### 1.3. Two-Facet GT Designs, Key Formulas, and Related SEMs

**Basic concepts.** To create a two-facet GT design, we will include occasions as an additional facet to produce a *persons × items × occasions* (*pio*) random-effects design. Within this design, each person responds to all items on all occasions. The partitioning of estimated variance at individual item and item-mean score levels is shown in Equations (7) and (8).

$$pio \text{ design : Individual item score level} : \hat{\sigma}^2_{Y_{pio}} = \hat{\sigma}^2_p + \hat{\sigma}^2_{pi} + \hat{\sigma}^2_{po} + \hat{\sigma}^2_{pio,e} + \hat{\sigma}^2_i + \hat{\sigma}^2_o + \hat{\sigma}^2_{io}, \tag{7}$$

$$pIO \text{ design : Item-mean score level} : \hat{\sigma}^2_{Y_{pIO}} = \hat{\sigma}^2_p + \frac{\hat{\sigma}^2_{pi}}{n'_i} + \frac{\hat{\sigma}^2_{po}}{n'_o} + \frac{\hat{\sigma}^2_{pio,e}}{n'_i n'_o}, \tag{8}$$

where $\hat{\sigma}^2$ = estimated variance component, $Y_{pio}$ = a score for a particular person on a given combination of item and occasion, $Y_{pIO}$ = the mean across all items and occasions for a particular person, $n'_i$ = number of items, and $n'_o$ = number of occasions.

Equation (7) reveals that the estimated variance in observed scores across persons, items, and occasions ($\hat{\sigma}^2_{Y_{pio}}$) is now partitioned into seven additive components representing

persons ($\hat{\sigma}_p^2$), inter-person differences in item and/or occasion scores ($\hat{\sigma}_{pi}^2$, $\hat{\sigma}_{po}^2$, $\hat{\sigma}_{pio,e}^2$), and absolute differences in item and/or occasion scores ($\hat{\sigma}_i^2$, $\hat{\sigma}_o^2$, $\hat{\sigma}_{io}^2$). When partitioning variance at the item-mean level in Equation (8), variance components for absolute differences again drop out because the mean for scores averaged over items and occasions is now a constant across persons.

**Indices of score consistency and agreement.** Formulas for G, global D, and cut-score specific D coefficients for the *pIO* design are provided in Equations (9)–(11).

$$\hat{G} \text{ coefficient for } pIO \text{ design } = \frac{\hat{\sigma}_p^2}{\hat{\sigma}_p^2 + \left( \frac{\hat{\sigma}_{pi}^2}{n_i'} + \frac{\hat{\sigma}_{po}^2}{n_o'} + \frac{\hat{\sigma}_{pio,e}^2}{n_i' n_o'} \right)} \tag{9}$$

$$\text{Global } \hat{D} \text{ coefficient for } pIO \text{ design } = \frac{\hat{\sigma}_p^2}{\hat{\sigma}_p^2 + \left( \frac{\hat{\sigma}_{pi}^2}{n_i'} + \frac{\hat{\sigma}_{po}^2}{n_o'} + \frac{\hat{\sigma}_{pio,e}^2}{n_i' n_o'} + \frac{\hat{\sigma}_i^2}{n_i'} + \frac{\hat{\sigma}_o^2}{n_o'} + \frac{\hat{\sigma}_{io}^2}{n_i' n_o'} \right)} \tag{10}$$

Cut-score specific $\hat{D}$ coefficient for $pIO$ design

$$= \frac{\hat{\sigma}_p^2 + \left[ \left( \bar{Y} - \text{Cut-score} \right)^2 - \hat{\sigma}_{\bar{Y}}^2 \right]}{\hat{\sigma}_p^2 + \left[ \left( \bar{Y} - \text{Cut-score} \right)^2 - \hat{\sigma}_{\bar{Y}}^2 \right] + \left( \frac{\hat{\sigma}_{pi}^2}{n_i'} + \frac{\hat{\sigma}_{po}^2}{n_o'} + \frac{\hat{\sigma}_{pio,e}^2}{n_i' n_o'} + \frac{\hat{\sigma}_i^2}{n_i'} + \frac{\hat{\sigma}_o^2}{n_o'} + \frac{\hat{\sigma}_{io}^2}{n_i' n_o'} \right)} \tag{11}$$

where $\hat{\sigma}_{\bar{Y}}^2 = \frac{\hat{\sigma}_p^2}{n_p'} + \frac{\hat{\sigma}_{pi}^2}{n_p' n_i'} + \frac{\hat{\sigma}_{po}^2}{n_p' n_o'} + \frac{\hat{\sigma}_{pio,e}^2}{n_p' n_i' n_o'} + \frac{\hat{\sigma}_i^2}{n_i'} + \frac{\hat{\sigma}_o^2}{n_o'} + \frac{\hat{\sigma}_{io}^2}{n_i' n_o'}$ and corrects for bias (see [29]).

Measurement error within Equation (9) for the G coefficient is now subdivided into three sources. $\hat{\sigma}_{pi}^2/n_i'$ reflects inter-person differences in the ordering of item scores (specific-factor error), $\hat{\sigma}_{po}^2/n_o'$ reflects inter-person differences in the ordering of occasion scores (transient error), and $\hat{\sigma}_{pio,e}^2/\left( n_i' \times n_o' \right)$ reflects inter-person differences in within-occasion "noise" (random-response error; see [24,38–40] for more extended discussions of these sources of measurement error). The global and cut-score specific D coefficients in Equations (10) and (11) also include three additional estimated variance components $\left( \hat{\sigma}_i^2, \hat{\sigma}_o^2, \hat{\sigma}_{io}^2 \right)$ to account for absolute differences among item and occasion scores. Values within parentheses for G and global D coefficients in Equations (9) and (10) again represent estimates of relative and absolute error. As before, when means across facet conditions are equal (i.e., $\hat{\sigma}_i^2 = \hat{\sigma}_o^2 = \hat{\sigma}_{io}^2 = 0$ here), relative and absolute error coincide, as do G and global D coefficients.

**SEM representation.** An SEM for the *pio* GT design based on administration of the same three items on two occasions is shown at the bottom of Figure 1. This model has orthogonal factors for person, each item, and each occasion. Occasion variances, item variances, and uniquenesses are, respectively, set equal, and all factor loadings are set equal to one. In total, the four variance components reflecting relative differences in scores ($\hat{\sigma}_p^2$, $\hat{\sigma}_{pi}^2$, $\hat{\sigma}_{po}^2$, $\hat{\sigma}_{pio,e}^2$) are directly estimated, but those for absolute differences ($\hat{\sigma}_i^2$, $\hat{\sigma}_o^2$, $\hat{\sigma}_{io}^2$) are excluded.

To estimate the remaining components, effect coding constraints are again imposed along with setting the sum of all item factor means and the sum of all occasion factor means equal to zero. When these restrictions are imposed, Jorgensen [14] noted that variance components for absolute differences in scores can be obtained using Equations (12)–(14).

$$\hat{\sigma}_i^2 = \frac{1}{n_i - 1} \sum_1^{n_i} (Item \ factor \ mean_i)^2 \tag{12}$$

$$\hat{\sigma}_o^2 = \frac{1}{n_o - 1} \sum_1^{n_o} (\text{Occasion factor mean}_o)^2 \tag{13}$$

$$\hat{\sigma}_{io}^2 = \frac{1}{(n_i \times n_o) - 1} \sum_{1}^{n_i \times n_o} (Intercept_{io})^2 \tag{14}$$

**Scale coarseness.** When considering results from GT analyses, we routinely assume that data are interval level in nature, meaning that equal differences in observed scores represent equal differences in the constructs being measured [1]. However, this assumption is not strictly met with most self-report measures due to scale coarseness effects resulting from limited numbers of response options and/or unequal underlying intervals between those options. To address this problem, Ark [13], Jorgensen [14] and Vispoel et al. [22] described how to conduct GT analyses on continuous latent response variable (CLRV) metrics using SEMs. The GT-SEMs are identical to those for observed scores described here except that estimation methods such as diagonally weighted least squares (DWLS) or paired maximum likelihood (PML) can be used to convert observed score results to CLRV metrics.

Vispoel at al. [22] used DWLS estimation for SEMs with delta parameterization to derive G and D coefficients for *pi* and *pio* designs with scores for indicators expressed on the same standardized scales. However, Jorgensen [14] noted that variability in means for latent variable indicators could be modeled to provide more informative D coefficients in CLRV analyses by using theta parameterization and constraining thresholds to be the same across all indicators. As a result, we adopted his approach when analyzing designs for CLRVs.

### 1.4. This Investigation

Our goal in the research reported here was to expand upon Jorgensen's [14] preliminary demonstration of ways to represent complete GT designs in SEM frameworks to encompass empirical data collected from respondents in live assessment settings who completed popular measures of personality, self-concept, and socially desirable responding on two occasions. We compared results from GT-SEM analyses using the *lavaan* package in R [41] to those obtained using the ANOVA-based package *GENOVA* [2], which remains one of the most comprehensive programs available for conducting traditional GT analyses (see, e.g., [20]). We further compared results from both packages to those obtained from *lavaan* using diagonally weighted least squares estimation to evaluate effects of scale coarseness and included Monte Carlo-based confidence intervals for variance components, proportions of measurement error, G coefficients, global D coefficients, and selected cut-score specific D coefficients within both the observed score and CLRV analyses.

## 2. Materials and Methods

### 2.1. Participants and Procedure

We collected data from three separate samples of college students from the University of Iowa who completed self-report inventories online using the Qualtrics platform on two occasions a week apart. Data collection was approved by the governing institutional review board (ID# 200809738) and all respondents gave informed consent before participating. Students within their respective samples completed all subscales from either the 100-item International Personality Item Pool Big Five Model Questionnaire (IPIP-BFM-100 [42]; n = 359, 69.58% female, 72.70% Caucasian; mean age = 23.80), Self-Description Questionnaire III (SDQ-III [43]; n = 427, 70.02% female, 78.69% Caucasian; mean age = 23.20), or Balanced Inventory of Desirable Responding (BIDR [44]; n = 595, 76.47% female, 77.31% Caucasian; mean age = 22.46). Inquiries about accessibility to the data should be directed to the first author.

### 2.2. Measures

**IPIP-BFM-100.** The IPIP-BFM-100 includes 100 items designed to measure the broad personality constructs associated with the Big Five model: Agreeableness, Conscientiousness, Emotional Stability, Extraversion, and Openness [42,45,46]. Each subscale has 20

items answered using a 5-point response metric (1 = Very Inaccurate, 5 = Very Accurate). Goldberg [42] reported alpha reliability estimates for the subscales ranging from 0.81 to 0.97 and exploratory factor analyses for self and peer ratings supporting the anticipated five-factor structure underlying item responses.

**SDQ-III.** The SDQ-III is a 136-item questionnaire intended for use with late adolescents and adults. It includes one subscale to measure overall self-esteem (General Self) and 12 additional ones to measure self-perceptions in the following areas: Emotional Stability, Honesty–Trustworthiness, Religious–Spiritual Values, Opposite-Sex Relations, Same-Sex Relations, Parental Relations, Physical Appearance, Physical Ability, General Academic Ability, Verbal Skills, Math Skills, and Problem-Solving Skills. Each scale has 10 or 12 items, equally balanced for negative and positive phrasing rated along an eight-point response metric (1 = Definitely False, 8 = Definitely True). Evidence reported by Marsh [43] and Byrne [47] in support of the reliability and construct validity of SDO-III subscale scores included alpha coefficients ranging from 0.76 to 0.95, median 1-month and 18-month test–retest coefficients, respectively, equaling 0.87 and 0.74, factor analyses verifying that each subscale measures a distinguishable construct, and logically consistent relationships of subscale scores with each other and external criterion measures.

**BIDR.** The BIDR (Version 6, [44]) has 40 items comprising two 20-item subscales that measure two dimensions of socially desirable responding: Impression Management and Self-Deceptive Enhancement. Items within each scale are equally balanced for positive and negative phrasing and rated along a 7-point response metric (1 = Not True, 4 = Somewhat True, 7 = Very True). After reversals are made to negatively phrased items, scores can remain on the original polytomous metric, be dichotomized to emphasize exaggeratedly desirable responses by rescoring extremely high responses (6 or 7) to equal 1 and other responses to equal 0, or be dichotomized to emphasize exaggeratedly undesirable responses by rescoring extremely low responses (1 or 2) to equal 0 and other responses to equal 1 [48–50]. We included all three approaches here. In practical settings, polytomous scores are more informative when Impression Management and Self-Deceptive Enhancement are treated as psychological traits, whereas dichotomized scores are frequently used to flag possible instances of faking good (i.e., exaggerated endorsement) or faking bad (i.e., exaggerated denial). Paulhus [44], Kilinc [48], Vispoel and Kim [49], Vispoel, Morris, and Clough [50], and Vispoel, Morris, and Sun [51] reported alpha coefficients for BIDR subscales ranging from 0.66 to 0.88, 1-week test–retest coefficients ranging from 0.71 to 0.88, and confirmation of anticipated patterns of convergent and discriminant validity coefficients with other measures.

## 3. Analyses

We analyzed fully crossed *pi* and *pio* random-facet GT designs for every subscale from each instrument using the ANOVA-based package *GENOVA* [2] and SEM-based *lavaan* package in R [41] with conventional least squares parameter estimates (labeled as expected mean squares in *GENOVA* and unweighted least squares (ULS) in *lavaan*). We repeated the *lavaan* analyses with robust diagonally weighted least squares (DWLS) estimates (WLSMV in R) using theta parameterization to evaluate the effects of scale coarseness and provide more informative D coefficients than those provided by delta parameterization. All SEMs were constrained in the ways described earlier to render indices reflecting both relative and absolute differences in scores. For each scale, design, and analysis, we estimated G coefficients, global D coefficients, cut-score specific D coefficients two standard deviations away from each scale's mean, and proportions of universe score and measurement error variance. We also derived 90% Monte Carlo-based confidence intervals for variance components, proportions of measurement error, G coefficients, global D coefficients, and cut-score specific D coefficients using the *semTools* package in R [26]. Within the reported GT analyses, $n_i'$ equals the number of items within a given subscale, and $n_o'$ equals one.

## 4. Results

### 4.1. Descriptive Statistics and Conventional Reliability Estimates

We provide means, standard deviations, alpha coefficients, and test–retest coefficients for all subscale scores from the IPIP-BFM-100, SDQ-III, and BIDR in Table 1. Across scales and inventories, alpha coefficients range from 0.691 to 0.956 ($M = 0.870$) and test–retest coefficients from 0.704 to 0.940 ($M = 0.857$). On average, alpha coefficients across occasions are highest for the IPIP-BFM-100 ($M = 0.918$), followed, respectively, by the SDQ-III ($M = 0.904$), BIDR with polytomous scores (M = 0.766), BIDR for dichotomous exaggerated endorsement scores ($M = 0.751$), and BIDR for dichotomous exaggerated denial scores ($M = 0.748$). Mean test–retest coefficients for IPIP-BFM-100, SDQ-III, BIDR (polytomous), BIDR (dichotomous exaggerated endorsement), and BIDR (dichotomous exaggerated denial) scores, respectively, equal 0.888, 0.889, 0.826, 0.748, and 0.707. Overall, these indices along with subscale means and standard deviations align well with those for college students from the studies previously cited.

**Table 1.** Descriptive Statistics and Conventional Reliability Estimates for IPIP-BFM-100, SDQ-III, and BIDR Scores.

| Instrument and Subscale | Number of Items | Index | | | | | | |
|---|---|---|---|---|---|---|---|---|
| | | Time 1 | | | Time 2 | | | |
| | | Mean Scale (Item) | SD Scale (Item) | Alpha | Mean Scale (Item) | SD Scale (Item) | Alpha | Test Retest |
| **IPIP-BFM-100** | | | | | | | | |
| Extraversion | 20 | 64.22 (3.21) | 13.72 (0.69) | 0.924 | 65.31 (3.27) | 14.08 (0.70) | 0.935 | 0.926 |
| Agreeableness | 20 | 80.37 (4.02) | 9.83 (0.49) | 0.883 | 79.98 (4.00) | 10.11 (0.51) | 0.904 | 0.853 |
| Conscientiousness | 20 | 73.70 (3.68) | 12.41 (0.62) | 0.914 | 73.44 (3.67) | 12.41 (0.62) | 0.924 | 0.896 |
| Emotional stability | 20 | 62.32 (3.12) | 15.09 (0.75) | 0.934 | 63.20 (3.16) | 15.13 (0.76) | 0.942 | 0.887 |
| Openness | 20 | 74.11 (3.71) | 11.14 (0.56) | 0.901 | 73.90 (3.69) | 11.45 (0.57) | 0.913 | 0.879 |
| Mean | 20 | 70.94 (3.55) | 12.44 (0.62) | 0.911 | 71.17 (3.56) | 12.63 (0.63) | 0.924 | 0.888 |
| **SDQ-III** | | | | | | | | |
| General self-esteem | 12 | 71.81 (5.98) | 16.33 (1.36) | 0.952 | 70.76 (5.90) | 16.62 (1.39) | 0.954 | 0.898 |
| Emotional stability | 10 | 51.04 (5.10) | 13.83 (1.38) | 0.910 | 51.71 (5.17) | 13.66 (1.37) | 0.913 | 0.880 |
| General academic skills | 10 | 58.56 (5.86) | 11.69 (1.17) | 0.911 | 58.21 (5.82) | 11.68 (1.17) | 0.914 | 0.862 |
| Verbal skills | 10 | 46.73 (4.67) | 18.03 (1.80) | 0.952 | 47.30 (4.73) | 17.22 (1.72) | 0.946 | 0.936 |
| Math skills | 10 | 56.80 (5.68) | 11.71 (1.17) | 0.866 | 56.70 (5.67) | 11.51 (1.15) | 0.869 | 0.888 |
| Problem-solving skills | 10 | 53.78 (5.38) | 10.05 (1.00) | 0.833 | 53.04 (5.30) | 10.73 (1.07) | 0.868 | 0.870 |
| Physical ability | 10 | 54.66 (5.47) | 16.74 (1.67) | 0.950 | 53.96 (5.40) | 16.71 (1.67) | 0.956 | 0.928 |
| Physical appearance | 10 | 50.43 (5.04) | 12.86 (1.29) | 0.916 | 50.40 (5.04) | 12.91 (1.29) | 0.924 | 0.918 |
| Opposite-sex relations | 10 | 55.24 (5.52) | 12.21 (1.22) | 0.882 | 55.04 (5.50) | 12.52 (1.25) | 0.902 | 0.882 |
| Same-sex relations | 10 | 57.16 (5.72) | 10.94 (1.09) | 0.849 | 56.30 (5.63) | 11.40 (1.14) | 0.880 | 0.843 |
| Parental relations | 10 | 61.87 (6.19) | 13.88 (1.39) | 0.923 | 61.77 (6.18) | 13.67 (1.37) | 0.930 | 0.897 |
| Honesty–trustworthiness | 12 | 74.42 (6.20) | 9.99 (0.83) | 0.773 | 73.28 (6.11) | 10.99 (0.92) | 0.836 | 0.813 |
| Religious–spiritual values | 12 | 59.09 (4.92) | 21.83 (1.82) | 0.949 | 59.09 (4.92) | 22.23 (1.85) | 0.954 | 0.940 |
| Mean | 10.46 | 57.81 (5.52) | 13.85 (1.32) | 0.897 | 57.50 (5.49) | 13.99 (1.34) | 0.911 | 0.889 |
| **BIDR Polytomous (1–7)** | | | | | | | | |
| Impression management | 20 | 80.62 (4.03) | 16.43 (0.82) | 0.766 | 81.33 (4.07) | 18.11 (0.91) | 0.826 | 0.830 |
| Self-deceptive enhancement | 20 | 82.18 (4.11) | 13.54 (0.68) | 0.717 | 84.32 (4.22) | 13.89 (0.69) | 0.754 | 0.822 |
| Mean | 20 | 81.40 (4.07) | 14.99 (0.75) | 0.742 | 82.82 (4.14) | 16.00 (0.80) | 0.790 | 0.826 |
| **BIDR Dichotomous (0 = 1–5; 1 = 6–7)** | | | | | | | | |
| Impression management | 20 | 6.74 (0.34) | 3.47 (0.17) | 0.726 | 14.52 (0.73) | 3.74 (0.19) | 0.808 | 0.783 |
| Self-deceptive enhancement | 20 | 5.27 (0.26) | 3.24 (0.16) | 0.691 | 16.42 (0.82) | 3.01 (0.15) | 0.779 | 0.712 |
| Mean | 20 | 6.00 (0.30) | 3.36 (0.17) | 0.708 | 15.47 (0.77) | 3.38 (0.17) | 0.793 | 0.748 |
| **BIDR Dichotomous (0 = 1–2; 1 = 3–7)** | | | | | | | | |
| Impression management | 20 | 13.90 (0.69) | 3.40 (0.17) | 0.735 | 6.39 (0.32) | 4.03 (0.20) | 0.797 | 0.711 |
| Self-deceptive enhancement | 20 | 15.66 (0.78) | 3.18 (0.16) | 0.728 | 5.33 (0.27) | 3.77 (0.19) | 0.733 | 0.704 |
| Mean | 20 | 14.78 (0.74) | 3.29 (0.16) | 0.731 | 5.86 (0.29) | 3.90 (0.19) | 0.765 | 0.707 |

### 4.2. Partitioning of Variance, G Coefficients, and D Coefficients on Observed Score Metrics

In Table 2, we provide variance components, G coefficients, global D coefficients, cut-score specific D coefficients two standard deviations away from the scale mean, and corresponding 90% confidence intervals for IPIP-BFM-100, SDQ-III, and BIDR observed

scores within the GT *pi* design analyses. Across subscales, *lavaan* and *GENOVA* results for G and D coefficients are identical to the three decimal places shown in the table, and variance components differ by no more than 0.005. Confidence intervals for all variance components fail to capture zero, thereby reflecting trustworthy effects. G coefficients (which mirror alpha coefficients reported in Table 2 for Occasion 1) range from 0.691 to 0.952 (*M* = 0.858), global D coefficients from 0.672 to 0.945 (*M* = 0.834) and cut-score specific D coefficients from 0.932 to 0.989 (*M* = 0.962). Confidence interval lower limits for G coefficients equal or exceed 0.830 in all instances except for subscales from the BIDR and the Honesty–Trustworthiness subscale from the SDQ-III. Lower limits for global D coefficients equal or exceed 0.804 except for subscales from the BIDR and the Honesty–Trustworthiness and Problem-Solving Skills subscales from the SDQ-III. Finally, lower limits for cut-score specific D coefficients two standard deviations away from the mean equal or exceed 0.915 for all scales across all instruments.

In Tables 3 and 4, we provide parallel indices for the GT *pio* designs plus additional variance components and partitioning of measurement error into three sources (specific-factor, transient, and random-response). Across subscales, *lavaan* and *GENOVA* results for G coefficients and proportions of measurement error are identical to the three decimal places shown in the tables; D coefficients differ by no more than 0.002; and variance components differ by no more than 0.013. Confidence intervals for all variance components and proportions of measurement error fail to capture zero except *o* variance components for most subscales across instruments, *io* variance components for the SDQ-III Problem-Solving Skills subscale and BIDR Self-Deceptive Enhancement dichotomous subscales, and both *po* variance components and proportions of transient error for all BIDR dichotomous subscales. Across instruments, *o* and *io* variance components are extremely low in magnitude (*M* for *o* = 0.0008; *M* for *io* = 0.0041), which makes sense given that means for occasions and for items across occasions were not expected to vary much over the one-week interval between administrations of the current trait-oriented measure.

Across subscales, G coefficients range from 0.592 to 0.915 (*M* = 0.795), global D coefficients from 0.561 to 0.909 (*M* = 0.774), cut-score specific D coefficients from 0.907 to 0.982 (*M* = 0.953), proportions of specific-factor error from 0.016 to 0.151 (*M* = 0.060), proportions of transient error from 0.036 to 0.150 (*M* = 0.076), and proportions of random-response error from 0.024 to 0.159 (*M* = 0.069). G and D coefficients and their corresponding confidence interval lower limits for the *pio* designs are less than those for the *pi* designs due to the inclusion of additional sources of measurement error. Confidence interval lower limits for G coefficients equal or exceed 0.807 in all instances except for the Agreeableness subscale from the IPIP-BFM-100; the Same-Sex Relations, Problem-Solving Skills, and Honesty–Trustworthiness subscales from the SDQ-III; and all subscales from the BIDR. Lower limits for global D coefficients equal or exceed 0.802 except for the Agreeableness subscale from the IPIP-BFM-100, the Verbal Skills, Same-Sex Relations, Problem-Solving Skills, and Honesty–Trustworthiness subscales from the SDQ-III, and all subscales from the BIDR. Lastly, lower limits for cut-score specific D coefficients two standard deviations away from the mean equal or exceed 0.874 for all subscales across all instruments.

**Table 2.** Variance Components, G coefficients, and D coefficients for GT Observed Score *pi* Designs.

| Instrument/ Subscale | Index | | | | | |
|---|---|---|---|---|---|---|
| | *p* | *pi,e* | *i* * | G | G-D | CS-D |
| **IPIP-BFM-100** | | | | | | |
| Extraversion | 0.435 (0.429, 0.441) | 0.719 (0.698, 0.739) | 0.173 (0.159, 0.193) | 0.924 (0.921, 0.926) | 0.907 (0.904, 0.910) | 0.981 (0.980, 0.982) |
| Agreeableness | 0.213 (0.207, 0.220) | 0.564 (0.543, 0.584) | 0.058 (0.052, 0.071) | 0.883 (0.878, 0.889) | 0.873 (0.866, 0.878) | 0.974 (0.973, 0.975) |
| Conscientiousness | 0.352 (0.346, 0.358) | 0.662 (0.641, 0.682) | 0.087 (0.078, 0.102) | 0.914 (0.911, 0.917) | 0.904 (0.900, 0.907) | 0.981 (0.980, 0.981) |
| Emotional stability | 0.532 (0.526, 0.538) | 0.752 (0.731, 0.772) | 0.142 (0.130, 0.161) | 0.934 (0.932, 0.936) | 0.922 (0.920, 0.925) | 0.984 (0.984, 0.985) |
| Openness | 0.280 (0.274, 0.286) | 0.614 (0.593, 0.634) | 0.067 (0.060, 0.081) | 0.901 (0.897, 0.905) | 0.891 (0.887, 0.895) | 0.978 (0.977, 0.979) |
| Mean | | | | 0.911 | 0.899 | 0.980 |
| **SDQ-III** | | | | | | |
| General self-esteem | 1.761 (1.751, 1.771) | 1.077 (1.052, 1.102) | 0.150 (0.134, 0.171) | 0.952 (0.950, 0.953) | 0.945 (0.944, 0.946) | 0.989 (0.989, 0.989) |
| Emotional stability | 1.740 (1.728, 1.752) | 1.729 (1.701, 1.757) | 0.606 (0.567, 0.650) | 0.910 (0.908, 0.911) | 0.882 (0.879, 0.884) | 0.976 (0.975, 0.976) |
| General academic skills | 1.244 (1.232, 1.256) | 1.219 (1.191, 1.247) | 0.250 (0.226, 0.279) | 0.911 (0.908, 0.913) | 0.894 (0.891, 0.897) | 0.978 (0.978, 0.979) |
| Verbal skills | 1.188 (1.176, 1.200) | 1.839 (1.811, 1.867) | 0.375 (0.345, 0.410) | 0.866 (0.863, 0.868) | 0.843 (0.840, 0.846) | 0.968 (0.967, 0.968) |
| Math skills | 3.096 (3.084, 3.108) | 1.562 (1.534, 1.590) | 0.235 (0.212, 0.264) | 0.952 (0.951, 0.953) | 0.945 (0.944, 0.946) | 0.989 (0.989, 0.989) |
| Problem-solving skills | 0.841 (0.829, 0.853) | 1.681 (1.653, 1.708) | 0.433 (0.400, 0.470) | 0.833 (0.830, 0.837) | 0.799 (0.794, 0.804) | 0.958 (0.957, 0.959) |
| Physical ability | 2.661 (2.649, 2.672) | 1.413 (1.386, 1.441) | 0.248 (0.224, 0.277) | 0.950 (0.949, 0.951) | 0.941 (0.940, 0.943) | 0.988 (0.988, 0.988) |
| Physical appearance | 1.515 (1.503, 1.527) | 1.381 (1.354, 1.409) | 0.665 (0.624, 0.711) | 0.916 (0.915, 0.918) | 0.881 (0.878, 0.884) | 0.975 (0.975, 0.976) |
| Opposite-sex relations | 1.316 (1.304, 1.327) | 1.753 (1.725, 1.781) | 0.403 (0.372, 0.440) | 0.882 (0.880, 0.885) | 0.859 (0.856, 0.862) | 0.971 (0.970, 0.972) |
| Same-sex relations | 1.015 (1.003, 1.027) | 1.807 (1.779, 1.834) | 0.609 (0.571, 0.654) | 0.849 (0.846, 0.852) | 0.808 (0.804, 0.812) | 0.960 (0.959, 0.960) |
| Parental relations | 1.780 (1.768, 1.791) | 1.483 (1.455, 1.511) | 0.383 (0.353, 0.419) | 0.923 (0.921, 0.925) | 0.905 (0.903, 0.907) | 0.981 (0.980, 0.981) |
| Honesty–trustworthiness | 0.536 (0.526, 0.545) | 1.884 (1.859, 1.909) | 0.519 (0.487, 0.557) | 0.773 (0.769, 0.778) | 0.728 (0.722, 0.733) | 0.942 (0.941, 0.943) |
| Religious–spiritual values | 3.138 (3.128, 3.148) | 2.038 (2.013, 2.064) | 0.328 (0.303, 0.358) | 0.949 (0.948, 0.949) | 0.941 (0.940, 0.942) | 0.988 (0.988, 0.988) |
| Mean | | | | 0.897 | 0.875 | 0.974 |
| **BIDR Polytomous (1–7)** | | | | | | |
| Impression management | 0.517 (0.512, 0.522) | 3.164 (3.148, 3.180) | 1.038 (1.008, 1.071) | 0.766 (0.764, 0.768) | 0.711 (0.708, 0.714) | 0.938 (0.937, 0.938) |
| Self-deceptive enhancement | 0.329 (0.324, 0.334) | 2.592 (2.576, 2.608) | 0.431 (0.412, 0.453) | 0.717 (0.714, 0.721) | 0.685 (0.681, 0.689) | 0.934 (0.933, 0.935) |
| Mean | | | | 0.742 | 0.698 | 0.936 |
| **BIDR Dichotomous (0 = 1–5; 1 = 6–7)** | | | | | | |
| Impression management | 0.022 (0.017, 0.027) | 0.165 (0.149, 0.181) | 0.039 (0.034, 0.047) | 0.726 (0.664, 0.773) | 0.682 (0.615, 0.731) | 0.932 (0.917, 0.943) |
| Self-deceptive enhancement | 0.018 (0.013, 0.023) | 0.162 (0.146, 0.178) | 0.014 (0.012, 0.020) | 0.691 (0.610, 0.749) | 0.672 (0.588, 0.729) | 0.933 (0.915, 0.944) |
| Mean | | | | 0.731 | 0.697 | 0.936 |
| **BIDR Dichotomous (0 = 1–2; 1 = 3–7)** | | | | | | |
| Impression management | 0.021 (0.016, 0.026) | 0.153 (0.137, 0.169) | 0.040 (0.036, 0.048) | 0.735 (0.671, 0.783) | 0.687 (0.618, 0.737) | 0.933 (0.917, 0.944) |
| Self-deceptive enhancement | 0.018 (0.014, 0.023) | 0.138 (0.122, 0.153) | 0.015 (0.013, 0.021) | 0.728 (0.652, 0.783) | 0.707 (0.627, 0.761) | 0.940 (0.922, 0.951) |
| Mean | | | | 0.708 | 0.677 | 0.932 |

*Note.* *p* = person, *pi,e* = person × item and other error, *i* = item, G = G coefficient, G-D = global D coefficient, CS-D = cut-score specific D coefficient. Table entries represent results obtained from *lavaan* in R. * Differences for *i* variance components between *lavaan* and *GENOVA* range between 0.002 and 0.005 (*M* = 0.003). All other indices between packages are identical to the three decimal places shown in the table. Values within parentheses are 90% confidence interval limits.

**Table 3.** Variance Components for GT Observed Score *pio* Designs.

| Instrument/ Subscale | Index | | | | | | |
|---|---|---|---|---|---|---|---|
| | *p* | *pi* | *po* | *pio,e* | *i* | *o* | *io* |
| **IPIP-BFM-100** | | | | | | | |
| Extraversion | 0.429 (0.424, 0.433) | 0.375 (0.354, 0.395) | 0.021 (0.014, 0.027) | 0.305 (0.280, 0.330) | 0.154 [a] (0.144, 0.166) | 0.001 (0.000, 0.003) | 0.001 (0.001, 0.004) |
| Agreeableness | 0.200 (0.195, 0.204) | 0.246 (0.226, 0.266) | 0.022 (0.016, 0.029) | 0.281 (0.256, 0.306) | 0.054 [a] (0.049, 0.062) | 0.000 (0.000, 0.001) | 0.001 (0.001, 0.003) |
| Conscientiousness | 0.329 (0.324, 0.333) | 0.324 (0.304, 0.344) | 0.025 (0.019, 0.031) | 0.300 (0.275, 0.325) | 0.081 [a] (0.074, 0.090) | 0.000 (0.000, 0.001) | 0.001 (0.001, 0.003) |
| Emotional stability | 0.489 (0.485, 0.494) | 0.343 (0.322, 0.363) | 0.046 (0.040, 0.053) | 0.368 (0.343, 0.393) | 0.118 [a] (0.110, 0.129) | 0.001 (0.000, 0.003) | 0.002 (0.002, 0.005) [a] |
| Openness | 0.264 (0.260, 0.269) | 0.319 (0.299, 0.339) | 0.025 (0.019, 0.032) | 0.274 (0.249, 0.299) | 0.052 [a] (0.047, 0.060) | 0.000 (0.000, 0.001) | 0.002 (0.002, 0.005) [a] |
| **SDQ-III** | | | | | | | |
| General self-esteem | 1.661 (1.654, 1.668) | 0.368 (0.343, 0.392) | 0.135 (0.125, 0.145) | 0.698 (0.668, 0.728) | 0.122 [a] (0.111, 0.135) | 0.004 (0.002, 0.007) [a] | 0.003 (0.002, 0.006) [a] |
| Emotional stability | 1.573 (1.565, 1.582) | 0.886 (0.860, 0.913) | 0.148 (0.137, 0.160) | 0.789 (0.756, 0.822) | 0.568 [a] (0.541, 0.597) | 0.002 (0.000, 0.005) [a] | 0.001 (0.001, 0.004) |
| General academic skills | 1.130 (1.121, 1.138) | 0.470 (0.443, 0.496) | 0.116 (0.104, 0.128) | 0.724 (0.691, 0.757) | 0.213 [a] (0.197, 0.232) | 0.001 (0.000, 0.002) [a] | 0.004 (0.003, 0.008) [a] |
| Verbal skills | 1.099 (1.090, 1.107) | 0.989 (0.962, 1.015) | 0.071 (0.059, 0.083) | 0.800 (0.767, 0.833) | 0.316 (0.297, 0.339) | 0.000 (0.000, 0.001) [a] | 0.003 (0.002, 0.007) [a] |
| Math skills | 2.840 (2.831, 2.848) | 0.657 (0.630, 0.683) | 0.111 (0.099, 0.123) | 0.919 (0.885, 0.952) | 0.205 [a] (0.190, 0.224) | 0.002 [a] (0.000, 0.004) | 0.002 [a] (0.002, 0.005) |
| Problem-solving skills | 0.868 (0.860, 0.877) | 0.695 (0.669, 0.722) | 0.052 (0.041, 0.064) | 0.903 (0.870, 0.936) | 0.427 [a] (0.404, 0.453) | 0.003 (0.001, 0.006) | 0.000 [a] (0.000, 0.002) |
| Physical ability | 2.542 (2.534, 2.551) | 0.518 (0.491, 0.544) | 0.122 (0.110, 0.133) | 0.809 (0.776, 0.842) | 0.177 [a] (0.162, 0.194) | 0.002 [a] (0.001, 0.006) | 0.005 [a] (0.004, 0.009) |
| Physical appearance | 1.454 (1.446, 1.463) | 0.697 (0.671, 0.724) | 0.073 (0.061, 0.085) | 0.628 (0.595, 0.661) | 0.633 [a] (0.605, 0.665) | 0.000 [a] (0.000, 0.001) | 0.003 [a] (0.002, 0.006) |
| Opposite-sex relations | 1.272 (1.264, 1.281) | 0.763 (0.736, 0.789) | 0.093 (0.081, 0.105) | 0.880 (0.846, 0.913) | 0.353 [a] (0.332, 0.377) | 0.000 [a] (0.000, 0.002) | 0.002 [a] (0.002, 0.006) |
| Same-sex relations | 0.984 (0.976, 0.993) | 0.670 (0.643, 0.697) | 0.095 (0.083, 0.107) | 1.014 (0.980, 1.047) | 0.567 [a] (0.540, 0.597) | 0.004 (0.001, 0.007) | 0.002 [a] (0.002, 0.005) |
| Parental relations | 1.634 (1.625, 1.642) | 0.687 (0.661, 0.714) | 0.126 (0.114, 0.137) | 0.709 (0.676, 0.742) | 0.328 [a] (0.307, 0.351) | 0.000 [a] (0.000, 0.001) | 0.003 [a] (0.002, 0.007) |
| Honesty–trustworthiness | 0.553 (0.546, 0.560) | 0.804 (0.780, 0.828) | 0.066 (0.056, 0.076) | 0.965 (0.935, 0.995) | 0.486 [a] (0.463, 0.511) | 0.005 [a] (0.002, 0.008) | 0.005 [a] (0.004, 0.009) |
| Religious–spiritual values | 3.084 (3.077, 3.091) | 0.993 (0.969, 1.018) | 0.123 (0.113, 0.133) | 0.964 (0.934, 0.994) | 0.276 [a] (0.260, 0.295) | 0.000 [a] (0.000, 0.001) | 0.003 [a] (0.003, 0.007) |
| **BIDR Polytomous (1–7)** | | | | | | | |
| Impression management | 0.539 (0.536, 0.542) | 1.574 (1.558, 1.589) | 0.058 (0.053, 0.063) | 1.431 (1.412, 1.450) | 0.942 [a] (0.922, 0.964) | 0.001 [a] (0.000, 0.002) | 0.007 [a] (0.006, 0.010) |
| Self-deceptive enhancement | 0.316 (0.312, 0.319) | 1.417 (1.401, 1.432) | 0.031 (0.026, 0.035) | 1.065 (1.046, 1.084) | 0.385 [a] (0.373, 0.400) | 0.006 [a] (0.004, 0.008) | 0.004 [a] (0.004, 0.006) |
| **BIDR Dichotomous (0 = 1–5; 1 = 6–7)** | | | | | | | |
| Impression management | 0.024 (0.020, 0.027) | 0.071 (0.056, 0.087) | 0.003 (−0.001, 0.008) | 0.090 (0.070, 0.109) | 0.034 [a] (0.031, 0.039) | 0.000 (0.000, 0.001) | 0.001 [a] (0.001, 0.002) |
| Self-deceptive enhancement | 0.018 (0.015, 0.022) | 0.069 (0.054, 0.085) | 0.005 (0.000, 0.010) | 0.090 (0.071, 0.109) | 0.013 (0.012, 0.017) | 0.000 (0.000, 0.000) | 0.000 (0.000, 0.001) |
| **BIDR Dichotomous (0 = 1–2; 1 = 3–7)** | | | | | | | |
| Impression management | 0.020 (0.016, 0.023) | 0.056 (0.040, 0.071) | 0.005 (0.000, 0.010) | 0.092 (0.072, 0.111) | 0.035 [a] (0.032, 0.040) | 0.000 (0.000, 0.001) | 0.001 [a] (0.001, 0.002) |
| Self-deceptive enhancement | 0.014 (0.011, 0.018) | 0.053 (0.037, 0.068) | 0.003 (−0.002, 0.008) | 0.076 (0.057, 0.095) | 0.012 (0.011, 0.016) | 0.001 (0.000, 0.002) | 0.000 (0.000, 0.002) |

*Note.* $p$ = person, $pi$ = person $\times$ item, $po$ = person $\times$ occasion, $pio$ = person $\times$ item $\times$ occasion and other error, $i$ = item, $o$ = occasion, $io$ = item $\times$ occasion. Table entries represent results obtained from *lavaan* in R. [a] Differences for these indices between *lavaan* and *GENOVA* range between −0.011 and 0.013 ($M$ = 0.001). All other indices between packages are identical to the three decimal places shown in the table. Values within parentheses are 90% confidence interval limits.

**Table 4.** G coefficients, D coefficients, and Partitioning of Variance for GT Observed Score *pIO* Designs.

| Instrument/ Subscale | Index | | | | | |
|---|---|---|---|---|---|---|
| | G | SFE | TE | RRE | G-D | CS-D |
| **IPIP-BFM-100** | | | | | | |
| Extraversion | 0.887 (0.875, 0.899) | 0.039 (0.037, 0.041) | 0.043 (0.030, 0.056) | 0.032 (0.029, 0.034) | 0.870 (0.858, 0.882) | 0.974 (0.971, 0.976) |
| Agreeableness | 0.804 (0.781, 0.827) | 0.050 (0.045, 0.054) | 0.090 (0.066, 0.115) | 0.057 (0.051, 0.062) | 0.794 [a] (0.771, 0.816) | 0.958 (0.954, 0.963) |
| Conscientiousness | 0.854 (0.839, 0.869) | 0.042 (0.039, 0.045) | 0.065 (0.049, 0.081) | 0.039 (0.036, 0.042) | 0.845 (0.830, 0.859) | 0.969 (0.966, 0.972) |
| Emotional stability | 0.857 (0.847, 0.867) | 0.030 (0.028, 0.032) | 0.081 (0.070, 0.092) | 0.032 (0.030, 0.034) | 0.846 [a] (0.836, 0.856) | 0.969 (0.967, 0.971) |
| Openness | 0.828 (0.810, 0.846) | 0.050 (0.047, 0.053) | 0.079 (0.060, 0.098) | 0.043 (0.039, 0.047) | 0.821 (0.803, 0.838) | 0.964 (0.960, 0.967) |
| Mean | 0.846 | 0.042 | 0.072 | 0.040 | 0.835 | 0.967 |
| **SDQ-III** | | | | | | |
| General self-esteem | 0.881 (0.876, 0.886) | 0.016 (0.015, 0.017) | 0.072 (0.067, 0.077) | 0.031 (0.030, 0.032) | 0.875 (0.869, 0.879) | 0.975 (0.974, 0.976) |
| Emotional stability | 0.833 (0.827, 0.838) | 0.047 (0.045, 0.048) | 0.079 (0.072, 0.085) | 0.042 (0.040, 0.044) | 0.807 (0.802, 0.813) | 0.960 (0.959, 0.961) |
| General academic skills | 0.828 (0.820, 0.835) | 0.034 (0.032, 0.036) | 0.085 (0.077, 0.093) | 0.053 (0.051, 0.056) | 0.814 (0.806, 0.822) | 0.962 (0.961, 0.964) |
| Verbal skills | 0.814 (0.807, 0.822) | 0.073 (0.071, 0.075) | 0.053 (0.044, 0.061) | 0.059 (0.057, 0.062) | 0.796 (0.788, 0.803) | 0.958 (0.956, 0.960) |
| Math skills | 0.914 (0.910, 0.917) | 0.021 (0.020, 0.022) | 0.036 (0.032, 0.039) | 0.030 (0.028, 0.031) | 0.907 (0.903, 0.911) | 0.981 (0.981, 0.982) |
| Problem-solving skills | 0.804 (0.794, 0.813) | 0.064 (0.062, 0.067) | 0.049 (0.038, 0.059) | 0.084 (0.080, 0.087) | 0.771 (0.761, 0.780) | 0.952 (0.950, 0.954) |
| Physical ability | 0.909 (0.905, 0.913) | 0.019 (0.018, 0.019) | 0.043 (0.039, 0.048) | 0.029 (0.028, 0.030) | 0.902 (0.898, 0.906) | 0.980 (0.979, 0.981) |
| Physical appearance | 0.876 (0.870, 0.883) | 0.042 (0.040, 0.044) | 0.044 (0.037, 0.051) | 0.038 (0.036, 0.040) | 0.844 (0.837, 0.850) | 0.968 (0.966, 0.969) |
| Opposite-sex relations | 0.832 (0.825, 0.839) | 0.050 (0.048, 0.052) | 0.061 (0.053, 0.068) | 0.058 (0.055, 0.060) | 0.813 (0.806, 0.819) | 0.962 (0.960, 0.963) |
| Same-sex relations | 0.789 (0.781, 0.797) | 0.054 (0.052, 0.056) | 0.076 (0.067, 0.085) | 0.081 (0.078, 0.084) | 0.752 (0.744, 0.760) | 0.948 (0.946, 0.950) |
| Parental relations | 0.860 (0.855, 0.866) | 0.036 (0.035, 0.038) | 0.066 (0.060, 0.072) | 0.037 (0.036, 0.039) | 0.846 (0.840, 0.851) | 0.969 (0.967, 0.970) |
| Honesty–trustworthiness | 0.722 (0.711, 0.733) | 0.088 (0.085, 0.090) | 0.086 (0.074, 0.098) | 0.105 (0.102, 0.108) | 0.681 (0.670, 0.691) | 0.932 (0.930, 0.935) |
| Religious–spiritual values | 0.915 (0.912, 0.918) | 0.025 (0.024, 0.025) | 0.037 (0.034, 0.039) | 0.024 (0.023, 0.025) | 0.909 (0.906, 0.911) | 0.982 (0.981, 0.982) |
| Mean | 0.844 | 0.044 | 0.060 | 0.052 | 0.824 | 0.964 |
| **BIDR Polytomous (1–7)** | | | | | | |
| Impression management | 0.721 (0.716, 0.727) | 0.105 (0.104, 0.106) | 0.078 (0.071, 0.084) | 0.096 (0.094, 0.097) | 0.678 [a] (0.672, 0.683) | 0.931 (0.930, 0.932) |
| Self-deceptive enhancement | 0.671 (0.662, 0.680) | 0.151 (0.149, 0.153) | 0.065 (0.055, 0.075) | 0.113 (0.111, 0.115) | 0.637 [a] (0.628, 0.646) | 0.923 (0.921, 0.925) |
| Mean | 0.696 | 0.128 | 0.071 | 0.104 | 0.657 | 0.927 |
| **BIDR Dichotomous (0 = 1–5; 1 = 6–7)** | | | | | | |
| Impression management | 0.674 (0.563, 0.798) | 0.100 (0.077, 0.125) | 0.099 (−0.043, 0.223) | 0.127 (0.097, 0.159) | 0.640 [b] (0.533, 0.752) | 0.924 (0.901, 0.948) |
| Self-deceptive enhancement | 0.592 (0.469, 0.729) | 0.112 (0.086, 0.142) | 0.150 (−0.010, 0.289) | 0.146 (0.112, 0.184) | 0.579 [a] (0.457, 0.708) | 0.914 (0.888, 0.940) |
| Mean | 0.633 | 0.106 | 0.124 | 0.136 | 0.610 | 0.919 |
| **BIDR Dichotomous (0 = 1–2; 1 = 3–7)** | | | | | | |
| Impression management | 0.620 (0.500, 0.753) | 0.088 (0.062, 0.115) | 0.149 (−0.004, 0.284) | 0.143 (0.110, 0.180) | 0.579 [a] (0.466, 0.698) | 0.910 (0.885, 0.935) |
| Self-deceptive enhancement | 0.593 (0.436, 0.773) | 0.110 (0.076, 0.149) | 0.138 (−0.074, 0.314) | 0.159 (0.115, 0.211) | 0.561 [a] (0.411, 0.725) | 0.907 (0.874, 0.941) |
| Mean | 0.606 | 0.099 | 0.144 | 0.151 | 0.570 | 0.909 |

*Note.* G = G coefficient, SFE = specific-factor error, TE = transient error, RRE = random-response error, G-D = global D coefficient, CS-D = cut-score specific D coefficient. Table entries represent results obtained from *lavaan* in R. Values within parentheses are 90% confidence interval limits. Differences with *GENOVA* are indicated by superscripts: [a] The *lavaan* result is 0.001 lower than in *GENOVA*; [b] The *lavaan* result is 0.002 lower than in *GENOVA*.

### 4.3. Partitioning of Variance, G coefficients, and D coefficients on CLRV Metrics

In Tables 5–7, we provide the same indices for CLRVs as those reported in Tables 2–4 for observed scores within the *pi* and *pio* designs based on WLSMV estimates from *lavaan*. For the *pi* design results within Table 5, G coefficients range from 0.756 to 0.976 ($M = 0.909$), global D coefficients from 0.726 to 0.969 ($M = 0.886$), and cut-score specific D coefficients from 0.943 to 0.994 ($M = 0.976$). In all instances, score consistency and agreement indices as well as their corresponding confidence interval lower limits exceed those from the observed score analyses. As was the case with observed scores, confidence intervals for all CLRV variance components fail to capture zero, again underscoring trustworthy effects. Minimum confidence interval lower limits for G, global D, and cut-score specific D coefficients for CLRVs, respectively, equal 0.733, 0.700, and 0.937 as compared to 0.691, 0.588, and 0.915 for observed scores.

For the CLRV *pio* design results in Table 7, G coefficients range from 0.684 to 0.923 ($M = 0.819$), global D coefficients from 0.653 to 0.917 ($M = 0.800$), cut-score specific D coefficients from 0.927 to 0.983 ($M = 0.959$), proportions of specific-factor error from 0.012 to 0.143 ($M = 0.050$), proportions of transient error from 0.041 to 0.189 ($M = 0.099$), and proportions of random-response error from 0.011 to 0.081 ($M = 0.032$). Confidence intervals for all variance components and proportions of measurement error fail to capture zero except *o* components for 19 of the 24 subscales and the *io* component for the SDQ-III's Problem-Solving Skills subscale. As was the case with observed scores, CLRV variance components for *o* ($M = 0.0024$) and *io* ($M = 0.0042$) are extremely low in magnitude in comparison to the other variance components. Differences in lower confidence interval limits between CRLVs and observed scores for G, global D, and cut-score specific D coefficients vary with subscale. Across the 24 subscales, CLRV lower confidence interval limits are greater than or equal to those for observed scores in 10 instances for G coefficients, 10 instances for global D coefficients, and 14 instances for cut-score specific D coefficients. Minimum lower limits for G, global D, and cut-score specific D coefficients, respectively, equal 0.637, 0.614, and 0.917 for CLRVs versus 0.436, 0.411, and 0.874 for observed scores.

In Table 8, we report differences between WLSMV and ULS SEMs in G coefficients, global D coefficients, cut-score specific D coefficients, and proportions of measurement error for all designs to further evaluate effects of scale coarseness. For the *pi* designs, differences between WLSMV and ULS G and D coefficients are greater for dichotomously scored BIDR scales than for those with five to eight response options, with differences across all scales being noticeably greater for G and global D coefficients than for D coefficients representing cut-scores two standard deviations away from the scale mean. However, even for scales with five to eight options, G and global D coefficients are uniformly higher for WLSMV than for ULS with differences ranging from 0.014 to 0.087 ($M = 0.034$) for G coefficients and from 0.014 to 0.095 ($M = 0.036$) for global D coefficients.

In the *pio* designs, differences between WLSMV and ULS in G and global D coefficients are lower than those in the *pi* designs and again markedly higher for dichotomous scales than for polytomous scales. For the polytomous scales, differences in G and global D coefficients on average are generally quite small ($Ms = 0.010$ and $0.011$), with the largest being for the Honesty–Trustworthiness subscale from the SDQ-III. The general pattern of differences in relative proportions of variance between WLSMV and ULS within each inventory is for relative proportions of universe score and transient error to increase and relative proportions of specific-factor and random-response error to decrease.

**Table 5.** Variance Components, G coefficients, and D coefficients for GT CLRV *pi* Designs.

| Instrument/ Subscale | Index | | | | | |
|---|---|---|---|---|---|---|
| | *p* | *pi,e* | *i* | G | G-D | CS-D |
| **IPIP-BFM-100** | | | | | | |
| Extraversion | 1.744 (1.548, 1.938) | 1.866 (1.816, 1.918) | 0.570 (0.520, 0.638) | 0.949 (0.943, 0.954) | 0.935 (0.927, 0.941) | 0.987 (0.985, 0.988) |
| Agreeableness | 0.545 (0.482, 0.610) | 0.816 (0.777, 0.856) | 0.105 (0.094, 0.125) | 0.930 (0.922, 0.937) | 0.922 (0.913, 0.929) | 0.984 (0.982, 0.986) |
| Conscientiousness | 0.723 (0.633, 0.814) | 0.925 (0.884, 0.967) | 0.155 (0.138, 0.179) | 0.940 (0.933, 0.946) | 0.930 (0.922, 0.937) | 0.986 (0.984, 0.987) |
| Emotional stability | 1.091 (0.958, 1.225) | 1.097 (1.058, 1.137) | 0.260 (0.236, 0.292) | 0.952 (0.946, 0.957) | 0.941 (0.934, 0.947) | 0.988 (0.987, 0.989) |
| Openness | 0.604 (0.532, 0.675) | 0.830 (0.791, 0.869) | 0.109 (0.094, 0.131) | 0.936 (0.928, 0.942) | 0.928 (0.919, 0.934) | 0.985 (0.984, 0.987) |
| Mean | | | | 0.941 | 0.931 | 0.986 |
| **SDQ-III** | | | | | | |
| General self-esteem | 1.928 (1.686, 2.172) | 0.747 (0.705, 0.789) | 0.156 (0.134, 0.184) | 0.969 (0.965, 0.972) | 0.962 (0.958, 0.966) | 0.992 (0.991, 0.993) |
| Emotional stability | 0.924 (0.817, 1.032) | 0.584 (0.557, 0.611) | 0.270 (0.237, 0.310) | 0.941 (0.934, 0.946) | 0.915 (0.905, 0.924) | 0.983 (0.980, 0.984) |
| General academic skills | 0.975 (0.851, 1.099) | 0.623 (0.592, 0.655) | 0.157 (0.136, 0.184) | 0.940 (0.932, 0.946) | 0.926 (0.916, 0.933) | 0.985 (0.983, 0.986) |
| Verbal skills | 0.665 (0.584, 0.745) | 0.698 (0.665, 0.731) | 0.168 (0.148, 0.193) | 0.905 (0.894, 0.914) | 0.885 (0.872, 0.895) | 0.976 (0.974, 0.979) |
| Math skills | 0.877 (0.770, 0.985) | 0.314 (0.301, 0.326) | 0.064 (0.054, 0.076) | 0.965 (0.961, 0.969) | 0.959 (0.953, 0.963) | 0.992 (0.991, 0.993) |
| Problem-solving skills | 0.483 (0.422, 0.545) | 0.725 (0.691, 0.759) | 0.194 (0.173, 0.219) | 0.870 (0.854, 0.882) | 0.840 (0.822, 0.855) | 0.967 (0.963, 0.970) |
| Physical ability | 1.723 (1.520, 1.927) | 0.616 (0.590, 0.641) | 0.158 (0.140, 0.181) | 0.965 (0.961, 0.969) | 0.957 (0.951, 0.961) | 0.991 (0.990, 0.992) |
| Physical appearance | 0.939 (0.828, 1.051) | 0.581 (0.555, 0.606) | 0.352 (0.321, 0.387) | 0.942 (0.935, 0.948) | 0.910 (0.899, 0.919) | 0.981 (0.979, 0.983) |
| Opposite-sex relations | 0.724 (0.640, 0.807) | 0.667 (0.637, 0.697) | 0.193 (0.171, 0.221) | 0.916 (0.906, 0.923) | 0.894 (0.882, 0.903) | 0.978 (0.976, 0.980) |
| Same-sex relations | 0.586 (0.508, 0.663) | 0.708 (0.675, 0.741) | 0.287 (0.256, 0.324) | 0.892 (0.879, 0.903) | 0.855 (0.837, 0.869) | 0.970 (0.966, 0.973) |
| Parental relations | 1.334 (1.156, 1.512) | 0.728 (0.693, 0.763) | 0.320 (0.287, 0.360) | 0.948 (0.941, 0.954) | 0.927 (0.916, 0.935) | 0.985 (0.983, 0.987) |
| Honesty–trustworthiness | 0.455 (0.403, 0.506) | 0.887 (0.850, 0.925) | 0.288 (0.259, 0.324) | 0.860 (0.846, 0.872) | 0.823 (0.804, 0.838) | 0.963 (0.959, 0.966) |
| Religious–spiritual values | 2.072 (1.853, 2.292) | 0.600 (0.577, 0.623) | 0.193 (0.167, 0.226) | 0.976 (0.974, 0.979) | 0.969 (0.965, 0.972) | 0.994 (0.993, 0.994) |
| Mean | | | | 0.930 | 0.909 | 0.981 |
| **BIDR Polytomous (1–7)** | | | | | | |
| Impression management | 0.227 (0.199, 0.256) | 0.962 (0.939, 0.986) | 0.351 (0.329, 0.377) | 0.825 (0.806, 0.841) | 0.776 (0.752, 0.795) | 0.952 (0.947, 0.957) |
| Self-deceptive enhancement | 0.133 (0.117, 0.149) | 0.857 (0.829, 0.885) | 0.149 (0.137, 0.164) | 0.756 (0.733, 0.776) | 0.726 (0.700, 0.747) | 0.943 (0.937, 0.947) |
| Mean | | | | 0.791 | 0.751 | 0.948 |
| **BIDR Dichotomous (0 = 1–5; 1 = 6–7)** | | | | | | |
| Impression management | 0.336 (0.288, 0.384) | 1.091 (1.083, 1.100) | 0.510 (0.476, 0.553) | 0.860 (0.841, 0.876) | 0.808 (0.782, 0.827) | 0.959 (0.953, 0.963) |
| Self-deceptive enhancement | 0.252 (0.216, 0.288) | 1.009 (1.003, 1.015) | 0.205 (0.185, 0.233) | 0.833 (0.810, 0.851) | 0.806 (0.780, 0.826) | 0.960 (0.954, 0.964) |
| Mean | | | | 0.847 | 0.807 | 0.959 |
| **BIDR Dichotomous (0 = 1–2; 1 = 3–7)** | | | | | | |
| Impression management | 0.381 (0.326, 0.436) | 1.145 (1.137, 1.154) | 0.539 (0.504, 0.585) | 0.869 (0.850, 0.884) | 0.819 (0.794, 0.838) | 0.962 (0.956, 0.966) |
| Self-deceptive enhancement | 0.329 (0.278, 0.381) | 1.005 (0.999, 1.011) | 0.221 (0.201, 0.250) | 0.868 (0.847, 0.884) | 0.843 (0.818, 0.861) | 0.968 (0.962, 0.971) |
| Mean | | | | 0.868 | 0.831 | 0.965 |

*Note.* CLRV = continuous latent response variable, *p* = person, *pi,e* = person × item and other error, *i* = item, G = G coefficient, G-D = global D coefficient, CS-D = cut-score specific D coefficient. Values within parentheses are 90% confidence interval limits.

**Table 6.** Variance Components for GT CLRV *pio* Designs.

| Instrument/ Subscale | Index | | | | | | |
|---|---|---|---|---|---|---|---|
| | *p* | *pi* | *po* | *pio,e* | *i* | *o* | *io* |
| **IPIP-BFM-100** | | | | | | | |
| Extraversion | 1.001 (0.880, 1.123) | 0.677 (0.635, 0.718) | 0.076 (0.063, 0.089) | 0.373 (0.356, 0.389) | 0.307 (0.279, 0.342) | 0.003 (0.001, 0.006) | 0.002 (0.002, 0.004) |
| Agreeableness | 0.657 (0.581, 0.733) | 0.617 (0.571, 0.664) | 0.088 (0.067, 0.109) | 0.357 (0.342, 0.373) | 0.127 (0.114, 0.149) | 0.000 (0.000, 0.002) | 0.002 (0.002, 0.004) |
| Conscientiousness | 1.038 (0.900, 1.177) | 0.846 (0.785, 0.909) | 0.106 (0.084, 0.128) | 0.474 (0.451, 0.496) | 0.222 (0.196, 0.257) | 0.000 (0.000, 0.002) | 0.002 (0.002, 0.005) |
| Emotional stability | 1.495 (1.296, 1.696) | 0.943 (0.882, 1.003) | 0.184 (0.149, 0.220) | 0.596 (0.570, 0.623) | 0.327 (0.297, 0.367) | 0.003 (0.000, 0.009) | 0.006 (0.005, 0.010) |
| Openness | 0.923 (0.803, 1.042) | 0.848 (0.786, 0.911) | 0.111 (0.088, 0.135) | 0.462 (0.441, 0.483) | 0.139 (0.120, 0.168) | 0.000 (0.000, 0.002) | 0.005 (0.005, 0.009) |
| **SDQ-III** | | | | | | | |
| General self-esteem | 0.654 (0.559, 0.748) | 0.106 (0.095, 0.117) | 0.068 (0.060, 0.077) | 0.172 (0.162, 0.182) | 0.041 (0.035, 0.050) | 0.002 (0.000, 0.004) | 0.001 (0.001, 0.003) |
| Emotional stability | 1.218 (1.067, 1.370) | 0.511 (0.473, 0.549) | 0.157 (0.135, 0.178) | 0.347 (0.332, 0.361) | 0.386 (0.339, 0.439) | 0.001 (0.000, 0.005) | 0.001 (0.001, 0.003) |
| General academic skills | 1.588 (1.397, 1.778) | 0.469 (0.429, 0.510) | 0.175 (0.146, 0.205) | 0.604 (0.581, 0.628) | 0.236 (0.203, 0.277) | 0.001 (0.000, 0.005) | 0.005 (0.004, 0.009) |
| Verbal skills | 0.649 (0.564, 0.734) | 0.477 (0.443, 0.511) | 0.057 (0.045, 0.069) | 0.257 (0.245, 0.269) | 0.152 (0.135, 0.174) | 0.000 (0.000, 0.001) | 0.001 (0.001, 0.003) |
| Math skills | 1.729 (1.514, 1.944) | 0.338 (0.313, 0.362) | 0.077 (0.065, 0.088) | 0.341 (0.330, 0.352) | 0.119 (0.101, 0.141) | 0.001 (0.000, 0.003) | 0.001 (0.001, 0.003) |
| Problem-solving skills | 0.661 (0.573, 0.749) | 0.445 (0.411, 0.480) | 0.059 (0.047, 0.072) | 0.485 (0.466, 0.505) | 0.262 (0.237, 0.292) | 0.002 (0.000, 0.005) | 0.000 (0.000, 0.002) |
| Physical ability | 2.053 (1.793, 2.311) | 0.343 (0.318, 0.368) | 0.117 (0.099, 0.135) | 0.399 (0.383, 0.415) | 0.142 (0.126, 0.164) | 0.002 (0.000, 0.006) | 0.004 (0.003, 0.007) |
| Physical appearance | 1.329 (1.161, 1.496) | 0.500 (0.466, 0.534) | 0.114 (0.100, 0.128) | 0.340 (0.325, 0.356) | 0.501 (0.459, 0.548) | 0.000 (0.000, 0.001) | 0.002 (0.002, 0.005) |
| Opposite-sex relations | 0.911 (0.798, 1.024) | 0.445 (0.411, 0.479) | 0.102 (0.086, 0.118) | 0.403 (0.387, 0.420) | 0.232 (0.206, 0.263) | 0.000 (0.000, 0.002) | 0.002 (0.001, 0.004) |
| Same-sex relations | 0.879 (0.766, 0.992) | 0.449 (0.411, 0.487) | 0.120 (0.100, 0.139) | 0.535 (0.516, 0.554) | 0.413 (0.371, 0.460) | 0.004 (0.001, 0.008) | 0.002 (0.002, 0.005) |
| Parental relations | 1.144 (0.984, 1.305) | 0.333 (0.304, 0.363) | 0.106 (0.087, 0.125) | 0.301 (0.288, 0.314) | 0.258 (0.232, 0.288) | 0.000 (0.000, 0.002) | 0.003 (0.002, 0.005) |
| Honesty–trustworthiness | 0.341 (0.292, 0.390) | 0.340 (0.304, 0.376) | 0.044 (0.035, 0.054) | 0.275 (0.252, 0.298) | 0.165 (0.145, 0.190) | 0.001 (0.000, 0.003) | 0.002 (0.002, 0.005) |
| Religious–spiritual values | 0.728 (0.615, 0.840) | 0.113 (0.101, 0.124) | 0.057 (0.051, 0.064) | 0.102 (0.095, 0.109) | 0.060 (0.051, 0.070) | 0.000 (0.000, 0.000) | 0.001 (0.001, 0.002) |
| **BIDR Polytomous (1–7)** | | | | | | | |
| Impression management | 0.310 (0.270, 0.349) | 0.820 (0.780, 0.860) | 0.048 (0.040, 0.055) | 0.434 (0.421, 0.447) | 0.438 (0.409, 0.471) | 0.000 (0.000, 0.001) | 0.003 (0.003, 0.005) |
| Self-deceptive enhancement | 0.163 (0.141, 0.184) | 0.679 (0.642, 0.716) | 0.022 (0.018, 0.026) | 0.385 (0.373, 0.397) | 0.173 (0.159, 0.191) | 0.003 (0.002, 0.004) | 0.002 (0.001, 0.003) |
| **BIDR Dichotomous (0 = 1–5; 1 = 6–7)** | | | | | | | |
| Impression management | 0.716 (0.614, 0.818) | 1.631 (1.605, 1.656) | 0.114 (0.090, 0.137) | 0.449 (0.408, 0.491) | 0.880 (0.823, 0.950) | 0.001 (0.000, 0.005) | 0.016 (0.012, 0.024) |
| Self-deceptive enhancement | 0.371 (0.316, 0.426) | 1.032 (1.004, 1.060) | 0.098 (0.077, 0.119) | 0.413 (0.380, 0.447) | 0.282 (0.257, 0.316) | 0.000 (0.000, 0.002) | 0.004 (0.003, 0.007) |
| **BIDR Dichotomous (0 = 1–2; 1 = 3–7)** | | | | | | | |
| Impression management | 0.809 (0.686, 0.932) | 1.723 (1.668, 1.778) | 0.217 (0.171, 0.263) | 0.746 (0.685, 0.808) | 1.090 (1.023, 1.174) | 0.015 (0.006, 0.027) | 0.023 (0.018, 0.035) |
| Self-deceptive enhancement | 0.373 (0.308, 0.439) | 0.933 (0.904, 0.963) | 0.086 (0.066, 0.106) | 0.397 (0.364, 0.431) | 0.270 (0.249, 0.302) | 0.017 (0.010, 0.026) | 0.004 (0.004, 0.008) |

*Note.* CLRV = continuous latent response variable, *p* = person, *pi* = person × item, *po* = person × occasion, *pio,e* = person × item × occasion and other error, *i* = item, *o* = occasion, *io* = item × occasion. Values within parentheses are 90% confidence interval limits.

**Table 7.** G coefficients, D coefficients, and Partitioning of Variance for GT CLRV *pio* Designs.

| Instrument/ Subscale | Index | | | | | |
|---|---|---|---|---|---|---|
| | G | SFE | TE | RRE | G-D | CS-D |
| **IPIP-BFM-100** | | | | | | |
| Extraversion | 0.886 (0.869, 0.901) | 0.030 (0.027, 0.033) | 0.067 (0.055, 0.081) | 0.016 (0.015, 0.019) | 0.872 (0.854, 0.888) | 0.974 (0.970, 0.977) |
| Agreeableness | 0.828 (0.798, 0.855) | 0.039 (0.035, 0.043) | 0.111 (0.085, 0.138) | 0.022 (0.020, 0.025) | 0.821 (0.790, 0.848) | 0.964 (0.958, 0.969) |
| Conscientiousness | 0.858 (0.834, 0.878) | 0.035 (0.032, 0.039) | 0.088 (0.069, 0.107) | 0.020 (0.017, 0.022) | 0.850 (0.825, 0.870) | 0.970 (0.965, 0.974) |
| Emotional stability | 0.851 (0.826, 0.874) | 0.027 (0.024, 0.030) | 0.105 (0.084, 0.127) | 0.017 (0.015, 0.019) | 0.842 (0.815, 0.865) | 0.968 (0.963, 0.973) |
| Openness | 0.839 (0.813, 0.862) | 0.039 (0.035, 0.043) | 0.101 (0.080, 0.123) | 0.021 (0.019, 0.024) | 0.834 (0.807, 0.856) | 0.966 (0.961, 0.971) |
| Mean | 0.852 | 0.034 | 0.094 | 0.019 | 0.844 | 0.968 |
| **SDQ-III** | | | | | | |
| General self-esteem | 0.877 (0.862, 0.890) | 0.012 (0.011, 0.013) | 0.092 (0.081, 0.104) | 0.019 (0.017, 0.022) | 0.871 (0.855, 0.884) | 0.974 (0.971, 0.977) |
| Emotional stability | 0.834 (0.813, 0.853) | 0.035 (0.032, 0.039) | 0.107 (0.092, 0.124) | 0.024 (0.021, 0.027) | 0.812 (0.788, 0.831) | 0.961 (0.956, 0.965) |
| General academic skills | 0.849 (0.829, 0.867) | 0.025 (0.023, 0.028) | 0.094 (0.079, 0.110) | 0.032 (0.029, 0.036) | 0.838 (0.816, 0.856) | 0.967 (0.963, 0.971) |
| Verbal skills | 0.833 (0.810, 0.853) | 0.061 (0.056, 0.068) | 0.073 (0.058, 0.089) | 0.033 (0.029, 0.037) | 0.817 (0.793, 0.837) | 0.963 (0.958, 0.967) |
| Math skills | 0.923 (0.913, 0.932) | 0.018 (0.016, 0.020) | 0.041 (0.035, 0.048) | 0.018 (0.016, 0.021) | 0.917 (0.905, 0.926) | 0.983 (0.981, 0.985) |
| Problem-solving skills | 0.813 (0.788, 0.834) | 0.055 (0.049, 0.061) | 0.073 (0.058, 0.089) | 0.060 (0.054, 0.067) | 0.786 (0.759, 0.807) | 0.956 (0.950, 0.960) |
| Physical ability | 0.915 (0.903, 0.925) | 0.015 (0.014, 0.017) | 0.052 (0.044, 0.061) | 0.018 (0.016, 0.020) | 0.908 (0.895, 0.919) | 0.981 (0.979, 0.984) |
| Physical appearance | 0.870 (0.854, 0.884) | 0.033 (0.030, 0.037) | 0.075 (0.064, 0.086) | 0.022 (0.020, 0.025) | 0.843 (0.823, 0.858) | 0.967 (0.963, 0.971) |
| Opposite-sex relations | 0.830 (0.808, 0.848) | 0.041 (0.037, 0.045) | 0.093 (0.079, 0.109) | 0.037 (0.033, 0.041) | 0.812 (0.789, 0.831) | 0.962 (0.957, 0.966) |
| Same-sex relations | 0.801 (0.777, 0.823) | 0.041 (0.037, 0.045) | 0.109 (0.092, 0.128) | 0.049 (0.044, 0.054) | 0.770 (0.743, 0.793) | 0.952 (0.946, 0.957) |
| Parental relations | 0.871 (0.852, 0.888) | 0.025 (0.022, 0.029) | 0.080 (0.067, 0.096) | 0.023 (0.020, 0.026) | 0.854 (0.833, 0.872) | 0.970 (0.966, 0.974) |
| Honesty–trustworthiness | 0.781 (0.756, 0.803) | 0.065 (0.060, 0.071) | 0.102 (0.083, 0.121) | 0.052 (0.048, 0.058) | 0.755 (0.728, 0.778) | 0.949 (0.943, 0.954) |
| Religious–spiritual values | 0.906 (0.895, 0.915) | 0.012 (0.011, 0.013) | 0.071 (0.064, 0.080) | 0.011 (0.009, 0.012) | 0.901 (0.889, 0.910) | 0.980 (0.978, 0.982) |
| Mean | 0.854 | 0.034 | 0.082 | 0.031 | 0.837 | 0.967 |
| **BIDR Polytomous (1–7)** | | | | | | |
| Impression management | 0.737 (0.709, 0.762) | 0.098 (0.090, 0.107) | 0.114 (0.095, 0.133) | 0.052 (0.047, 0.058) | 0.700 (0.670, 0.725) | 0.937 (0.930, 0.942) |
| Self-deceptive enhancement | 0.684 (0.652, 0.712) | 0.143 (0.132, 0.155) | 0.092 (0.075, 0.109) | 0.081 (0.074, 0.089) | 0.653 (0.620, 0.681) | 0.927 (0.920, 0.933) |
| Mean | 0.711 | 0.120 | 0.103 | 0.066 | 0.676 | 0.932 |
| **BIDR Dichotomous (0 = 1–5; 1 = 6–7)** | | | | | | |
| Impression management | 0.767 (0.731, 0.798) | 0.087 (0.078, 0.098) | 0.122 (0.096, 0.150) | 0.024 (0.021, 0.028) | 0.731 (0.694, 0.762) | 0.944 (0.935, 0.950) |
| Self-deceptive enhancement | 0.685 (0.637, 0.728) | 0.095 (0.086, 0.107) | 0.181 (0.143, 0.222) | 0.038 (0.034, 0.044) | 0.667 (0.619, 0.709) | 0.932 (0.921, 0.940) |
| Mean | 0.726 | 0.091 | 0.152 | 0.031 | 0.699 | 0.938 |
| **BIDR Dichotomous (0 = 1–2; 1 = 3–7)** | | | | | | |
| Impression management | 0.704 (0.656, 0.747) | 0.075 (0.067, 0.084) | 0.189 (0.149, 0.230) | 0.032 (0.028, 0.037) | 0.663 (0.614, 0.707) | 0.929 (0.917, 0.938) |
| Self-deceptive enhancement | 0.710 (0.656, 0.756) | 0.089 (0.078, 0.102) | 0.164 (0.124, 0.207) | 0.038 (0.033, 0.044) | 0.671 (0.614, 0.718) | 0.930 (0.917, 0.941) |
| Mean | 0.707 | 0.082 | 0.176 | 0.035 | 0.667 | 0.929 |

*Note.* CLRV = continuous latent response variable, G = G coefficient, SFE = specific-factor error, TE = transient error, RRE = random-response error, G-D = global D coefficient, CS-D = cut-score specific D coefficient. Values within parentheses are 90% confidence interval limits.

**Table 8.** Differences between GT WLSMV and ULS Analyses in Score Consistency, Agreement, and Measurement Error.

| Instrument/ Subscale | Number of Items | Number of Scale Points | *pi* Design (WLSMV-ULS) | | | *pio* Design (WLSMV-ULS) | | | | | |
|---|---|---|---|---|---|---|---|---|---|---|---|
| | | | G | G-D | CS-D | G | SFE | TE | RRE | G-D | CS-D |
| **IPIP-BFM-100** | | | | | | | | | | | |
| Extraversion | 20 | 5 | 0.026 | 0.028 | 0.006 | −0.001 | −0.009 | 0.024 | −0.015 | 0.002 | 0.000 |
| Agreeableness | 20 | 5 | 0.047 | 0.049 | 0.010 | 0.024 | −0.011 | 0.020 | −0.034 | 0.027 | 0.005 |
| Conscientiousness | 20 | 5 | 0.026 | 0.027 | 0.005 | 0.004 | −0.007 | 0.023 | −0.019 | 0.005 | 0.001 |
| Emotional stability | 20 | 5 | 0.018 | 0.019 | 0.004 | −0.006 | −0.003 | 0.024 | −0.015 | −0.005 | −0.001 |
| Openness | 20 | 5 | 0.035 | 0.036 | 0.007 | 0.011 | −0.011 | 0.022 | −0.022 | 0.012 | 0.003 |
| Mean | 20 | 5 | 0.030 | 0.032 | 0.006 | 0.007 | −0.008 | 0.023 | −0.021 | 0.008 | 0.002 |
| **SDQ-III** | | | | | | | | | | | |
| General self-esteem | 12 | 8 | 0.017 | 0.017 | 0.003 | −0.004 | −0.004 | 0.020 | −0.012 | −0.003 | −0.001 |
| Emotional stability | 10 | 8 | 0.031 | 0.034 | 0.007 | 0.001 | −0.012 | 0.029 | −0.018 | 0.004 | 0.001 |
| General academic skills | 10 | 8 | 0.029 | 0.031 | 0.006 | 0.021 | −0.009 | 0.009 | −0.021 | 0.023 | 0.005 |
| Verbal skills | 10 | 8 | 0.039 | 0.042 | 0.009 | 0.018 | −0.012 | 0.020 | −0.026 | 0.021 | 0.004 |
| Math skills | 10 | 8 | 0.014 | 0.014 | 0.003 | 0.009 | −0.003 | 0.005 | −0.011 | 0.009 | 0.002 |
| Problem-solving skills | 10 | 8 | 0.036 | 0.041 | 0.009 | 0.009 | −0.010 | 0.024 | −0.024 | 0.014 | 0.003 |
| Physical ability | 10 | 8 | 0.016 | 0.016 | 0.003 | 0.006 | −0.003 | 0.009 | −0.011 | 0.006 | 0.001 |
| Physical appearance | 10 | 8 | 0.025 | 0.029 | 0.006 | −0.006 | −0.009 | 0.030 | −0.016 | −0.001 | 0.000 |
| Opposite-sex relations | 10 | 8 | 0.033 | 0.035 | 0.007 | −0.002 | −0.009 | 0.032 | −0.021 | −0.001 | 0.000 |
| Same-sex relations | 10 | 8 | 0.043 | 0.047 | 0.010 | 0.012 | −0.013 | 0.033 | −0.033 | 0.017 | 0.004 |
| Parental relations | 10 | 8 | 0.025 | 0.022 | 0.004 | 0.011 | −0.011 | 0.014 | −0.014 | 0.009 | 0.002 |
| Honesty–trustworthiness | 12 | 8 | 0.087 | 0.095 | 0.021 | 0.059 | −0.023 | 0.016 | −0.053 | 0.074 | 0.017 |
| Religious–spiritual values | 12 | 8 | 0.028 | 0.028 | 0.006 | −0.009 | −0.013 | 0.035 | −0.013 | −0.008 | −0.002 |
| Mean | 10.46 | 8 | 0.033 | 0.035 | 0.007 | 0.010 | −0.007 | 0.018 | −0.020 | 0.011 | 0.003 |
| **BIDR Polytomous (1–7)** | | | | | | | | | | | |
| Impression management | 20 | 7 | 0.060 | 0.065 | 0.015 | 0.016 | −0.008 | 0.036 | −0.044 | 0.022 | 0.005 |
| Self-deceptive enhancement | 20 | 7 | 0.039 | 0.041 | 0.009 | 0.013 | −0.008 | 0.027 | −0.032 | 0.016 | 0.004 |
| Mean | 20 | 7 | 0.049 | 0.053 | 0.012 | 0.015 | −0.008 | 0.031 | −0.038 | 0.019 | 0.005 |
| **BIDR Dichotomous (0 = 1–5; 1 = 6–7)** | | | | | | | | | | | |
| Impression management | 20 | 2 | 0.134 | 0.125 | 0.027 | 0.093 | −0.013 | 0.023 | −0.103 | 0.091 | 0.019 |
| Self-deceptive enhancement | 20 | 2 | 0.142 | 0.134 | 0.027 | 0.093 | −0.017 | 0.031 | −0.108 | 0.088 | 0.018 |
| Mean | 20 | 2 | 0.138 | 0.130 | 0.027 | 0.093 | −0.015 | 0.027 | −0.105 | 0.089 | 0.018 |
| **BIDR Dichotomous (0 = 1–2; 1 = 3–7)** | | | | | | | | | | | |
| Impression management | 20 | 2 | 0.135 | 0.132 | 0.028 | 0.084 | −0.013 | 0.039 | −0.111 | 0.084 | 0.019 |
| Self-deceptive enhancement | 20 | 2 | 0.140 | 0.136 | 0.028 | 0.117 | −0.022 | 0.026 | −0.121 | 0.110 | 0.023 |
| Mean | 20 | 2 | 0.137 | 0.134 | 0.028 | 0.101 | −0.017 | 0.033 | −0.116 | 0.097 | 0.021 |

*Note.* CLRV = continuous latent response variable, WLSMV = robust diagonally weighted least squares, ULS = unweighted least squares, G = G coefficient, G-D = global D coefficient, CS-D = cut-score specific D coefficient, SFE = specific-factor error, TE = transient error, RRE = random-response error.

## 5. Discussion

### 5.1. Overview

Our goal in the present study was to illustrate recently developed techniques for expanding GT analyses within SEM frameworks more thoroughly than in previous studies by including measures assessing a broad range of psychological traits taken in live assessment settings, deriving indices relevant to both norm- and criterion-referenced interpretations of scores, constructing confidence intervals for a variety of key GT indices, and taking the effects of scale coarseness into account. While doing so, we analyzed results for 24 scales from widely administered inventories assessing multiple dimensions of personality, self-concept, and socially desirable responding with item scale metrics having from two to eight response categories. When considered collectively, the present results highlight the effectiveness of SEMs in replicating results from ANOVA models, the importance of taking multiple sources of measurement error into account, the value of Jorgensen's procedures for deriving GT-based dependability coefficients and Monte Carlo-based confidence intervals for key indices, and the benefits of conducting GT analyses on both observed score and CLRV metrics to gauge scale coarseness effects.

### 5.2. Sources of Measurement Error

Across the analyses for observed scores, mean proportions of explained variance for specific-factor, transient, and random-response error equaled 0.060, 0.076, and 0.069, respectively. These findings are highly consistent with those from previous studies in underscoring how reliability is routinely overestimated in single-occasion research studies by failing to take all relevant sources of measurement error into account [14,16,18–24,32–34,38,39,52–55]. The omission of such effects within single-occasion studies, in turn, can lead to the substantial overestimation of reliability and corresponding underestimation of relations between latent constructs when those indices are used to correct correlation coefficients for measurement error (see, e.g., [15,32,38,54]). Such findings emphasize the inherent limitations of single-occasion studies and the importance of using multi-occasion data to better represent reliability and validity of scores from measures of psychological traits.

### 5.3. Dependability Coefficients

Applications of SEMs to derive variance components for persons, sources of relative measurement error, and corresponding G coefficients in the published research literature date back to Marcoulides [10]. In Raykov and Marcoulides's [11] follow-up to that study, the authors cited an unpublished paper by Marcoulides [56] in which they alluded to using SEMs in deriving variance components for absolute error. However, they provided no further details about the procedures. Ark [13] later speculated that a Q method could be used to derive each variance component for absolute error separately but acknowledged that this procedure would be cumbersome and of limited utility. More recently, Jorgensen [14] used a small set of contrived data to demonstrate simpler methods for deriving variance components for absolute error using indices embedded within the same SEMs for one- and two-facet GT designs used by Marcoulides, Raykov, and other researchers (see, e.g., [20–22]).

When we applied Jorgensen's [14] procedures here to data obtained from three separate samples of respondents in live settings, results for G coefficients, global D coefficients, cut-score specific D coefficients, and proportions of measurement error within the *pi* and *pio* designs varied by no more than 0.002 from ones obtained from the ANOVA-based GT package *GENOVA*. These results coupled with those from Jorgensen's original study confirm that SEMs provide a viable option for doing complete GT analyses while offering additional benefits that traditional ANOVA-based analyses rarely provide.

### 5.4. Confidence Intervals for Parameters within GT Analyses

One such benefit demonstrated here was to derive Monte Carlo-based confidence intervals for all reported variance components, proportions of measurement error, and indices of score

consistency and agreement. Cronbach et al. [57] and others (see, e.g., [30]) have long emphasized the importance of gauging sampling variability in GT parameter estimates but methods to do so are unavailable within most ANOVA-based packages or limited to procedures based on restrictive assumptions. In contrast, the *semTools* package in R [26] readily allows for the derivation of more widely applicable Monte Carlo-based intervals at any desired level of confidence for all GT indices considered here simply by adding a few lines of code to link commands within *semTools* to *lavaan* (see our online Supplementary Materials).

Although hypothesis testing is not part of traditional GT analyses, confidence intervals for variance components can serve a similar function when evaluating effects for persons, sources of measurement error, and differences in absolute levels of scores by noting whether zero or other targeted values fall within the limits of the interval. Our 90% confidence intervals for variance components often captured zero for occasion effects and sometimes for item by occasion interaction and transient error effects, whereas G and D coefficients had interval limits no lower than 0.411 across all scales, though some scales clearly yielded much more reliable results than others. On the observed score metric within the *pio* designs, confidence interval lower limits for both G and global D coefficients exceeded 0.80 for most subscales from the IPIP-BFM-100 (4 out of 5) and SDQ-III (9 out of 13), but not for either subscale from the BIDR across scoring methods. Overall, these results make sense because the psychological traits we assessed were expected to remain stable over the one-week interval between administrations, whereas item means, universe scores, and measurement error effects were expected to vary among respondents as well as within and across scales.

*5.5. Effects of Scale Coarseness*

Another unique benefit of GT-SEMs illustrated here was to use WLSMV estimates in *lavaan* to transform binary- and ordinal-level observed scores to CLRV metrics. Although we do not advocate substituting CLRV indices directly for those representing observed scores, we find such indices useful in gauging the effects of scale coarseness on reliability and in disattenuating correlation coefficients simultaneously for measurement error and scale coarseness. Because differences between observed scores and CLRVs in consistency and agreement should diminish as scale options increase, indices for CLRVs can serve as upper bounds for improvements that might be gained by increasing response options [14,22]. In essence, doing GT on CLRV metrics serves a similar function as *n'* value changes within G and D coefficient formulas by informing ways that assessment procedures might be improved.

To evaluate possible scale coarseness effects here, we intentionally included measures that varied in number of response categories. As expected, differences between G and global D coefficients were more pronounced for BIDR dichotomous scales compared to scales that included five to eight options. In general, these results support use of polytomous BIDR scores when measuring individual differences in Impression Management and Self-Deceptive Enhancement but do not preclude use of dichotomous scores for detecting faking (see, e.g., [48,58,59]). However, even with scales having five or more response options, we observed noticeable differences in score consistency and agreement between WLSMV and ULS estimation in some instances.

One such instance that encompassed nearly all subscales with five or more response options was within the *pi* designs in which mean differences between WLSMV and ULS equaled 0.034 for G coefficients and 0.036 for global D coefficients. These results are likely due in part to confounding of universe score and transient error variance in the *pi* designs. For example, within the corresponding *pio* designs that separate out transient error effects, mean differences between WLSMV and ULS dropped to 0.010 for G coefficients and 0.011 for global D coefficients. The greatest differences for polytomous scales were for Honesty–Trustworthiness that had the lowest standard deviations, alpha coefficients, and test–retest coefficient among SDQ-III scales.

Another factor that may be responsible for some differences between WLSMV and ULS for multi-option scales was restricting factor loadings and uniquenesses to be the

same across items within the *pi* and *pio* GT-SEM designs. In recent studies of GT, in which models with equal and varying unstandardized factor loadings and/or uniquenesses have been compared (i.e., congeneric versus essential tau-equivalent relationships), reliability is typically higher for the less restricted models, which in turn may further reduce differences between WLSMV- and ULS-based indices of consistency and agreement (see, e.g., [19,23,60]. In comparison to G and global D coefficients, cut-score specific D coefficients two standard deviations away from the scale mean were higher and varied much less across estimation methods. These results underscore that classification decisions made from extreme cut-scores can be highly congruent even when overall score consistency is relatively low and dichotomous score distributions are highly skewed.

### 5.6. Additional Advantages of GT-SEMs and Further Research

Although not demonstrated explicitly here, SEMs have additional benefits over traditional ANOVA models that merit comment and future exploration. One recent extension just mentioned is to model congeneric rather than essential tau-equivalent relationships between indicators and underlying factors. As is the case when comparing conventional single-occasion omega to alpha coefficients that, respectively, reflect congeneric versus essential tau-equivalent relationships [61,62], G coefficients are generally higher for congeneric than for essential tau-equivalent factor models (see, e.g., [18,23,60]). The offsetting drawback to modeling congeneric relationships within GT designs is that generalization is restricted to items and occasions sharing the same characteristics as those sampled in contrast to the broader domains from which they were drawn [61].

Vispoel, Xu, and Schneider [60] further showed that, despite differences in the labeling of indices, GT congeneric and latent state–trait orthogonal method models [63,64] are equivalent under certain conditions, thereby providing a useful bridge between the two theories. Researchers also have demonstrated that models within both frameworks can be extended to account for variance due to item phrasing effects (negative, positive, or both) and allow for partitioning of variance at both total score and individual item levels (see, e.g., [16,17,60]). However, a limitation of the GT congeneric models used in these studies was that they were not configured to yield global and cut-score specific D coefficients, thereby highlighting an important area for further investigation.

Other noteworthy recent extensions of GT-SEMs are to represent multivariate [16,54,55,65] and bifactor designs [16,18,19,65]. When analyzing multivariate GT designs, variance components for individual subscale scores are the same as those obtained from univariate designs, but those for composite scores are functions of the variances of subscale scores that comprise them, the covariances among those subscale scores, and the weighting of each subscale in forming the composite (see, e.g., [16,30,54,55,57]). Multivariate GT designs are useful in providing a clearer mapping of content within the global domain represented by subscale and item scores and in producing more appropriate and typically higher indices of score consistency and agreement for composite scores than would a direct univariate analysis of composite scores that ignores subscale representation and interrelationships [18,54,55,65]. Multivariate GT designs would be analyzed as SEMs with individual subscales represented in the same ways as in univariate designs but allowing person and measurement error factors (when appropriate) to covary across subscales (see [16,22,54,55,65]). Multivariate GT designs also allow for calculation of correlation coefficients between pairs of subscale scores corrected for all associated sources of measurement error (see, e.g., [54,55]).

Bifactor GT designs bear similarities to both univariate and multivariate GT designs but further partition universe score variance into general factor effects representing variance common to all items across subscales and independent group factor effects representing additional systematic variance specific to each subscale. Score consistency indices in bifactor models are expanded beyond those in univariate and multivariate GT designs to reflect proportions of variance accounted for by just general factor effects, just group factor effects, and general and group factor effects combined. Such partitioning provides

a useful means to investigating score dimensionality and value added when reporting subscale in addition to composite scores (see, e.g., [65–68]). Bifactor SEMs would include a general factor linked to all items and orthogonal group factors linked to items within each subscale plus additional factors for sources of measurement error in multi-facet GT designs (see [18,19,65] for further details).

Although research in representing GT multivariate and bifactor designs within SEM frameworks is very limited, the techniques illustrated here for traditional univariate GT designs can be extended to those designs to derive G, global D, and cut-score specific D coefficients at composite and subscale score levels; reference results to either observed score or CLRV metrics; and yield Monte Carlo-based confidence intervals for key parameters of interest. As with univariate congeneric SEMs, congeneric multivariate and bifactor designs can produce G coefficients, but methods for obtaining corresponding global and cut-score specific D coefficients remain an important area for further development.

## 6. Summary and Conclusions

The results reported here coupled with those from previous studies provide compelling evidence that SEMs can reproduce all key indices from GT ANOVA models for one- and two-facet designs while yielding Monte Carlo-based confidence intervals for those indices and referencing results to either observed score or CLRV metrics. Emerging research also suggests that the techniques described here for univariate GT-SEM analyses can be extended to multivariate and bifactor GT-SEMs to provide additional insights into the nature of assessment domains, create more appropriate indices of score consistency and agreement for composite scores, and further partition universe score variance into independent components reflecting general and group factor effects. To aid readers in applying the GT techniques demonstrated here, we provide examples of all illustrated SEM analyses using the *lavaan* [41] and *semTools* [26] packages in R and examples of all illustrated ANOVA analyses using the *GENOVA* package [2]. Additional guidelines for analyzing and applying these and more complex traditional univariate GT ANOVA-based designs are provided by Vispoel, Xu, and Schneider [24] using the *gtheory* package in R. Related guidelines and illustrations, using the *lavaan* package in R, are provided by Vispoel, Lee, and Hong [54,55,65] for analyzing multivariate GT-SEM designs and by Vispoel, Lee, et al. [18,19,65] for analyzing bifactor GT-SEM designs. We hope readers find these resources valuable in applying and extending GT-SEM procedures to their own data.

**Supplementary Materials:** The following supporting information can be downloaded at: https://www.mdpi.com/article/10.3390/psych5020019/s1, Table S1: Example Data Structure for GENOVA *p* × *i* design—Occasion 1; Table S2: Example Data Structure for GENOVA *p* × *i* × *o* design; Table S3: Variance Components, G coefficients, and D coefficients for GT *pi* Observed Score Design; Table S4: Variance Components, G coefficients, and D coefficients for GT *pi* CLRV Design; Table S5: G coefficients, D coefficients, and Partitioning of Variance for GT *pio* Observed Score Design; Table S6: G coefficients, D coefficients, and Partitioning of Variance for GT *pio* CLRV Design; Table S7: Formulas for Persons by Items Generalizability Theory Designs; Table S8: Formulas for Persons by Items by Occasions Generalizability Theory Designs.

**Author Contributions:** Conceptualization, W.P.V., H.L., T.C. and H.H.; methodology, W.P.V.; formal analysis, H.L., T.C. and W.P.V.; investigation, W.P.V.; resources, W.P.V.; data curation, W.P.V. and H.L.; writing—original draft preparation, W.P.V.; writing—review and editing, W.P.V., H.L., T.C. and H.H.; visualization, W.P.V., H.H., T.C. and H.L.; supervision, W.P.V.; project administration, W.P.V.; funding acquisition, W.P.V.; creation of online supplement, T.C., W.P.V. and H.L. All authors have read and agreed to the published version of the manuscript.

**Funding:** This project received no external funding but did receive internal research assistant support from the Iowa Measurement Research Foundation.

**Institutional Review Board Statement:** This study was conducted in accordance with the Declaration of Helsinki and approved by the Institutional Review Board of the University of Iowa (ID# 200809738).

**Informed Consent Statement:** Informed consent was obtained from all subjects involved in the study.

**Data Availability Statement:** This study was not preregistered and inquiries about accessibility to the data should be forwarded to the lead author.

**Conflicts of Interest:** The authors declare no conflict of interest.

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
