# Peer review of "Using Structural Equation Modeling to Reproduce and Extend ANOVA-Based Generalizability Theory Analyses for Psychological Assessments"

_psych, doi:10.3390/psych5020019_

Round 1

Reviewer 1 Report

This is a nice illustration of the many recent advances made in estimating G- and D-coefficients, and the many advantages available when using SEM to do so.   I have only minor comments:

Line 41 lists the R package gtheory among the software that "us[es] ANOVA-based procedures", which is inaccurate.  The gtheory package utilizes the lme4 package to fit a linear mixed model.  It may also be worth distinguishing earlier between a model (e.g., ANOVA vs. LMM) and an estimator of model parameters (e.g., mean-squares estimation of variance components in the ANOVA procedures vs. (RE)ML estimation in the LMM framework), since Line 72 indicates gtheory uses REML.

Line 56: "draught" should be "fraught".

Line 71: "produced by the anova package in R using ANOVA..." is misleading.  There is no anova package; rather, anova() is a function in the stats package.  It seem sufficient to simply state "calculated in R using ANOVA..."

Eqs. 1 and 2 would be easier for readers to understand if the model were introduced (defining each component of Y_pi) before the corresponding variance components.  The first word of "Item" is italicized in Eq. 1, and there is no space after the colon in Eq. 2.

Lines 153 and 159 refer to "consistency" when talking about absolute error.  Isn't "agreement" a more appropriate term in this context?  I am thinking of the distinction as described by McGraw & Wong (1996), where consistency referred specifically to relative error.

Line 351 (also 383, 392, 412, 488 ...) states "Confidence intervals for all indices fail to capture zero, thereby reflecting trustworthy effects in all cases."  Reliability/generalizability of zero is a pretty low bar for trustworthy.  While I do not endorse fixed cutoffs as guidelines for how large a coefficient is "reliable enough," certainly there is some consensus that values < .5 or .6 are unacceptable (given nearly / less than half of variance in a composite is noise).  All the CIs reported in Table 2 still meet such a criterion, although a few G-coef CIs include values < .7 (which is another common lower-bound benchmark), and Table 4 has many more such lower limits.  Again, I don't subscribe to a particular benchmark, but I am concerned about giving less familiar readers a false impression that merely > 0 is sufficient.

Monte Carlo CIs were referenced frequently, but only a software implementation in semTools was cited.  An article for interested readers would be appropriate to cite (e.g., the one from the semTools::monteCarloCI help page: https://doi.org/10.1080/19312458.2012.679848).

Line 502 might continue by describing the CLRV-scale indices as being in the same spirit of substituting n' in a D study, to learn how measurements can be improved.

Line 557: "Multivariance" should be "Multivariate"

Author Response

Responses to Reviewer 1 for the manuscript entitled “Using Structural Equation Modeling to Reproduce and Extend ANOVA-based Generalizability Theory Analyses for Psychological Assessments”

This is a nice illustration of the many recent advances made in estimating G- and D-coefficients, and the many advantages available when using SEM to do so.   I have only minor comments:

We thank the reviewer for this favorable appraisal of our submission and have done our best to address any remaining concerns. All comments and our responses to then follow.

Line 41 lists the R package gtheory among the software that "us[es] ANOVA-based procedures", which is inaccurate.  The gtheory package utilizes the lme4 package to fit a linear mixed model.  It may also be worth distinguishing earlier between a model (e.g., ANOVA vs. LMM) and an estimator of model parameters (e.g., mean-squares estimation of variance components in the ANOVA procedures vs. (RE)ML estimation in the LMM framework), since Line 72 indicates gtheory uses REML.

We appreciate the reviewer for reminding us about this distinction and addressed it by rewriting the sentence in question to exclude gtheory and added Footnote 1 and corresponding citations/references for clarification as shown below.

Current lines 43-48

Applications of GT rely heavily on variance component estimates traditionally obtained using ANOVA-based procedures within software packages catered specifically to applications of GT such as GENOVA [2], urGENOVA [3], and EduG [4] or from variance component programs within popular statistical packages such as SPSS, SAS, STATA, R, MATLAB, and Minitab (see, e.g., [5]).1

Lines 666-670

 1The computational framework for GT analyses also can be represented within linear mixed models (see, e.g., Jiang, 2018; Jiang et al., 2020). For example, in contrast to the other GT programs listed here, the gtheory package (Moore, 2016) in R uses the lme4 package (Bates et al., 2022) to fit a linear mixed model to the data. Within the gtheory package, restricted maximum likelihood (REML) estimates are used to derive variance components rather than conventional expected mean squares. REML estimation also is an option available when using the variance component estimation programs in SPSS, SAS, STATA, MATLAB, and Minitab.

Bates, D., Maechler, M., Bolker, B., & Walker, S. lme4: Linear mixed-effects models using 'Eigen' and S4. R package version 1.1-31. Retrieved from https://cran.r-project.org/web/packages/lme4/lme4.pdf

Jiang, Z. (2018). Using the linear mixed-effect model framework to estimate generalizability variance components in R: A lme4 package application. Methodology: European Journal of Research Methods for the Behavioral and Social Sciences, 14(3), 133–142. https://doi.org/10.1027/1614-2241/a000149

Jiang. Z., Raymond, M., Shi, D., & DiStefano, C. (2020). Using a linear mixed-effect model framework to estimate multivariate generalizability theory parameters in R. Behavior Research Methods, 52, 2383-2393. https://doi.org/10.3758/s13428-020-01399-z

Line 56: "draught" should be "fraught".

Good catch! We made the correction on current line 61 as highlighted below.

Part of the reason for such omissions was that derivation of variance components for absolute differences in scores was often considered unwieldy and fraught with technical difficulties due, for example, to presumptions that the data matrices analyzed needed to be transposed to treat facet conditions as participants and participants as facet conditions [10].

Line 71: "produced by the anova package in R using ANOVA..." is misleading.  There is no anova package; rather, anova() is a function in the stats package.  It seem sufficient to simply state "calculated in R using ANOVA..."

Thanks again for noticing this. We corrected as follows on lines 73-78 of the revised manuscript.

When applying his procedures to that dataset with fully crossed one- and two-facet GT-SEM designs using the lavaan SEM package in R and maximum likelihood parameter estimates, he obtained generalizability (G or Er2) and dependability (D or F) coefficients that varied by more than 0.003 from those produced by the anova() function in R using ANOVA mean square (MS) estimates and the gtheory package in R using restricted maximum likelihood (REML) estimates.

Eqs. 1 and 2 would be easier for readers to understand if the model were introduced (defining each component of Y_pi) before the corresponding variance components.  The first word of "Item" is italicized in Eq. 1, and there is no space after the colon in Eq. 2.

We addressed these issues by adding a couple sentences after Basic concepts starting on current line 122 and making the corrections noted in Equations 1 and 2.

Basic concepts. Within a persons × items (pi) random-effects GT design, persons and items are fully crossed, allowing the observed score for a particular person and item to be decomposed into person, item, and residual effects. The associated variance of each effect is called a variance component. Equations 1 and 2 show how estimated variances for item and item mean scores are partitioned within the persons × items (pi) random-effects GT design.

Lines 153 and 159 refer to "consistency" when talking about absolute error.  Isn't "agreement" a more appropriate term in this context?  I am thinking of the distinction as described by McGraw & Wong (1996), where consistency referred specifically to relative error.

Kane, M. T., & Brennan, R. L. (1980). Agreement coefficients as indices of dependability for domain-referenced tests. Applied Psychological Measurement4(1), 105-126. https://doi.org/10.1177/014662168000400111

Subheading on lines 148 and 209.

Indices of score consistency and agreement.

Sentence on Lines 407-408.

In all instances, score consistency and agreement indices as well as their corresponding confidence interval lower limits exceed those from the observed score analyses.

Sentence on Lines 504-506.

One such benefit demonstrated here was to derive Monte Carlo-based confidence intervals for all reported variance components, proportions of measurement error, and indices of score consistency and agreement.

Sentence on Lines 547-549.

However, even with scales having five or more response options, we observed noticeable differences in score consistency and agreement between WLSMV and ULS estimation in some instances.

Sentence on Lines 561-566.

In recent studies of GT in which models with equal and varying unstandardized factor loadings and/or uniquenesses have been compared (i.e., congeneric versus essential tau-equivalent relationships) reliability is typically higher for the less restricted models, which in turn may further reduce differences between WLSMV and ULS based indices of consistency and agreement (see, e.g., [13,22,60].

Sentence on Lines 596-601.

Multivariate GT designs are useful in providing a clearer mapping of content within the global domain represented by subscale and item scores and in producing more appropriate and typically higher indices of score consistency and agreement for composite scores than would a direct univariate analysis of composite scores that ignores subscale representation and interrelationships [17,54].

Sentence on Lines 630-635.

Emerging research also suggests that the techniques described here for univariate GT-SEMs analyses can be extended to multivariate and bifactor GT-SEMs to provide additional insights into the nature of assessment domains, create more appropriate indices of score consistency and agreement for composite scores, and further partition universe score variance into independent components reflecting general and group factor effects.

Changes to the following headings.

Line 354.

4.2 Partitioning of Variance, G coefficients, and D coefficients on Observed Score Metrics

Line 404.

4.3 Partitioning of Variance, G coefficients, and D coefficients on CLRV Metrics

Changes to the following table titles.

Line 365.

Table 2. Variance components, G coefficients, and D coefficients for GT Observed Score pi Designs

Line 388.

Table 4. G coefficients, D coefficients, and Partitioning of Variance for GT Observed Score pIO Designs

Line 414.

Table 5. Variance Components, G coefficients, and D coefficients for GT CLRV pi Designs

Line 422.

Table 7. G coefficients, D coefficients, and Partitioning of Variance for GT CLRV pio Designs

Line 443.

Table 8. Differences between GT WLSMV and ULS Analyses in Score Consistency, Agreement, and Measurement Error

Line 351 (also 383, 392, 412, 488 ...) states "Confidence intervals for all indices fail to capture zero, thereby reflecting trustworthy effects in all cases."  Reliability/generalizability of zero is a pretty low bar for trustworthy.  While I do not endorse fixed cutoffs as guidelines for how large a coefficient is "reliable enough," certainly there is some consensus that values < .5 or .6 are unacceptable (given nearly / less than half of variance in a composite is noise).  All the CIs reported in Table 2 still meet such a criterion, although a few G-coef CIs include values < .7 (which is another common lower-bound benchmark), and Table 4 has many more such lower limits.  Again, I don't subscribe to a particular benchmark, but I am concerned about giving less familiar readers a false impression that merely > 0 is sufficient.

We agree and should have been more precise in identifying indices for which zero would be a reasonable basis for comparison. To this end, we treated zero as reasonable targets within confidence intervals for variance component and proportions of measurements error but described confidence intervals for G and D coefficients in a more descriptive way. Defining acceptable levels of reliability was not our focus, and we realize that such cut-points are arbitrary and highly context dependent. Please consider our highlighted changes below collectively in response to your concern.

Lines 358-364 (pi design observed score analyses).

Confidence intervals for all variance components fail to capture zero, thereby reflecting trustworthy effects. G coefficients (which mirror alpha coefficients reported in Table 2 for Occasion 1) range from 0.691 to 0.952 (M = 0.858), global D coefficients from 0.672 to 0.945 (M = 0.834) and cut-score specific D coefficients from 0.932 to 0.989 (M = 0.962). Confidence interval lower limits for G coefficients equal or exceed 0.830 in all instances except for subscales from the BIDR and the Honesty-Trustworthiness subscale from the SDQ-III. Lower limits for global D coefficients equal or exceed 0.804 except for subscales from the BIDR and the Honesty-Trustworthiness and Problem-Solving Skills subscales from the SDQ-III. Finally, lower limits for cut-score specific D coefficients two standard deviations away from the mean equal or exceed 0.915 for all scales across all instruments.

Lines 371-403 (pio design observed score analyses).

In Tables 3 and 4, we provide parallel indices for the GT pio designs plus additional variance components and partitioning of measurement error into three sources (specific-factor, transient, and random-response). Across subscales, lavaan and GENOVA results for G coefficients and proportions of measurement error are identical to the three decimal places shown in the tables, D coefficients differ by no more than 0.002, and variance components by no more than 0.013. Confidence intervals for all variance components and proportions of measurement error fail to capture zero except o variance components for most subscales across instruments, io variance components the SDQ-III Problem-Solving subscale and BIDR Self-Deceptive Enhancement dichotomous subscales, po variance components for all BIDR dichotomous scales and proportions of transient error for the BIDR exaggerated denial subscales. Across instruments, the o and io variance components are extremely low in magnitude (M for o = 0.0008; M for io = 0.0041), which makes sense given that means for occasions and for items across occasions were not expected to vary much over the one-week interval between administrations of the current trait-oriented measures.

Across subscales, G coefficients range from 0.592 to 0.915 (M = 0.795), global D coefficients from 0.561 to 0.907 (M = 0.774), cut-score specific D coefficients from 0.909 to 0.982 (M = 0.953), proportions of specific-factor error from 0.016 to 0.151 (M = 0.060), proportions of transient error from 0.036 to 0.150 (M = 0.076), and proportions of random-response error from 0.024 to 0.159 (M = 0.069). G and D coefficients and their corresponding confidence interval lower limits for the pio designs are less than those for the pi designs due to inclusion of additional sources of measurement error. The confidence interval lower limits for G coefficients equal or exceed 0.807 in all instances except for the Agreeableness subscale from the IPIP-BFM-100; the Same-Sex Relations, Problem-Solving Skills, and Honesty-Trustworthiness subscales from the SDQ-III; and all subscales from the BIDR. Lower limits for global D coefficients equal or exceed 0.802 except for the Agreeableness subscale from the IPIP-BFM-100; the Verbal Skills, Same-Sex Relations, Problem-Solving Skills, and Honesty-Trustworthiness subscales from the SDQ-III, and all subscales from the BIDR.  Lastly, the lower limits for cut-score specific D coefficients two standard deviations away from the mean equal or exceed 0.874 for all subscales across all instruments.

Lines 405-436 (pi design CLRV analyses).

In Tables 5-7, we provide the same indices for CLRVs as those reported in Table 2-4 for observed scores within the pi and pio designs based on WLSMV estimates from lavaan. For the pi design results within Table 5, G coefficients range from 0.756 to 0.976 (M = 0.909), global D coefficients from 0.726 to 0.969 (M = 0.886) and cut-score specific D coefficients from 0.943 to 0.994 (M = 0.976). In all instances, score consistency and agreement indices as well as corresponding confidence interval lower limits exceed those from the observed score analyses. As was the case with observed scores, confidence intervals for all CLRV variance components fail to capture zero, again underscoring trustworthy effects. Minimum confidence interval lower limits for G, global D, and cut-score specific D coefficients for CLRVs respectively equal 0.756, 0.700, and 0.937 compared to 0.691, 0.588, and 0.915 for observed scores.

            For the CLRV pio design results in Table 7, G coefficients range from 0.684 to 0.923 (M = 0.819), global D coefficients from 0.653 to 0.917 (M = 0.800), cut-score specific D coefficients from 0.927 to 0.983 (M = 0.959), proportions of specific-factor error from 0.012 to 0.143 (M = 0.050), proportions of transient error from 0.041 to 0.189 (M = 0.099), and proportions of random-response error from 0.011 to 0.081 (M = 0.032). Confidence intervals for all variance components and proportions of measurement error fail to capture zero except o components for 19 of the 24 subscales and the io component for the SDQ-III’s Problem-Solving Skills subscale.. As was the case with observed scores, CLRV variance components for o (M = 0.0024) and io (M = 0.0042) are extremely low in magnitude in comparison to other variance components. Differences in lower confidence interval limits between CRLVs and observed scores for G, global D, and cut-score specific D coefficients vary with subscale. Across the 24 subscales, CLRV lower confidence interval limits are greater than or equal to those for observed scores in 10 instances for G-coefficients, 10 instances for global D coefficients, and 14 instances for cut-score specific D coefficients. Minimum lower limits for G, global D, and cut-score specific D coefficients respectively equal 0.637, 0.614, and 0.917 for CLRVs versus 0.436, 0.411, and 0.874 for observed scores.

Discussion Section Lines 514-528.

Although hypothesis testing is not part of traditional GT analyses, confidence intervals for variance components can serve a similar function when evaluating effects for persons, sources of measurement error, and differences in absolute levels of scores by noting whether zero or other targeted values fall within the limits of the interval. Our 90% confidence intervals for variance components often captured zero for occasion and sometimes for item by occasion interaction and transient error effects, whereas G and D coefficients had lower limits far greater than zero across all scales, though some scales clearly yielded much more reliable results than others. On the observed score metric within the pio designs, confidence interval lower limits for both G and global D coefficients exceeded 0.80 for most subscales from the IPIP=BFM-100 (4 out of 5) and SDQ-III (9 out of 13), but not for either subscale from the BIDR across scoring methods. Overall, these results make sense because the psychological traits we assessed were expected to remain stable over the one-week interval between administrations, whereas item means, universe scores, and measurement error effects were expected to vary among respondents as well as within and across scales.

Monte Carlo CIs were referenced frequently, but only a software implementation in semTools was cited.  An article for interested readers would be appropriate to cite (e.g., the one from the semTools::monteCarloCI help page: https://doi.org/10.1080/19312458.2012.679848).

We appreciate this suggestion and address it by adding Footnote 2 and the citations/references shown below.

Lines 671-673.

2The semTools package also can create Monte Carlo based confidence intervals from packages outside of R if an asymptotic sampling covariance (ACOV) matrix of variance-component parameters is available. More detailed information about Monte Carlo based confidence intervals can be found in Buckland (1984), Jiang, et al. (2022), and Preacher and Selig (2012).

Added references

Buckland, S. T. (1984). Monte Carlo confidence intervals. Biometrics, 40(3), 811-817. https://doi.org/10.2307/2530926

Jiang, Z., Raymond, M., DiStefano, C., Shi, D., Liu, R., & Sun, J. (2022). A Monte Carlo study of confidence interval methods for generalizability coefficient. Educational and Psychological Measurement, 82(4), 705–718. https://doi.org/10.1177/00131644211033899

Preacher, K. J., & Selig, J. P. (2012). Advantages of Monte Carlo confidence intervals for indirect effects. Communication Methods and Measures, 6(2),77-98. https://doi.org/10.1080/19312458.2012.679848

Line 502 might continue by describing the CLRV-scale indices as being in the same spirit of substituting n' in a D study, to learn how measurements can be improved.

This is an excellent suggestion that we handled it by adding the following sentence starting on line 538.

In essence, doing GT analyses on CLRV metrics serves a similar function as n’ value changes within G and D coefficient formulas by informing ways that assessment procedures might be improved.

Line 557: "Multivariance" should be "Multivariate"

We make this correction on current line 596.

Multivariate GT designs are useful in providing a clearer mapping of content within the global domain represented by subscale and item scores and in producing more appropriate and typically higher indices of score consistency for composite scores than would a direct univariate analysis of composite scores that ignores subscale representation and interrelationships [17,54].

We again thank the reviewer for this excellent feedback and hope that we adequately addressed the issues raised.

Reviewer 2 Report

This manuscript is quite well-written and clear. The only suggestion that I have for improvement is that I usually think of GT in the context of multilevel modeling, whereas the paper seems to use fixed-effects SEM models (while using random-effects GT). I would like to see the authors more clearly describe the relationship between the present work and multilevel implementations of GT (e.g., the gtheory package in R). 

Author Response

This manuscript is quite well-written and clear. The only suggestion that I have for improvement is that I usually think of GT in the context of multilevel modeling, whereas the paper seems to use fixed-effects SEM models (while using random-effects GT). I would like to see the authors more clearly describe the relationship between the present work and multilevel implementations of GT (e.g., the gtheory package in R). 

We thank the reviewer for the positive reaction to our submission. Our response to the suggestion mentioned entailed changes to lines 38-43, adding Footnote 1, and including additional citations/references as described below.

Lines 43-48

Applications of GT rely heavily on variance component estimates traditionally obtained using ANOVA-based procedures within software packages catered specifically to applications of GT such as GENOVA [2], urGENOVA [3], and EduG [4] or from variance component programs within popular statistical packages such as SPSS, SAS, STATA, R, MATLAB, and Minitab (see, e.g., [6]).1

Lines 666-670

Footnote 1

 1The computational framework for GT analyses also can be represented within linear mixed models (see, e.g., Jiang, 2018; Jiang et al., 2020). For example, in contrast to the other GT programs listed here, the gtheory package (Moore, 2016) in R uses the lme4 package (Bates et al., 2022) to fit a linear mixed model to the data. Within the gtheory package, restricted maximum likelihood (REML) estimates are used to derive variance components rather than conventional expected mean squares. REML estimation also is an option available when using the variance component estimation programs in SPSS, SAS, STATA, MATLAB, and MINITAB.

Added references.

Bates, D., Maechler, M., Bolker, B., & Walker, S. lme4: Linear mixed-effects models using 'Eigen' and S4. R package version 1.1-31. Retrieved from https://cran.r-project.org/web/packages/lme4/lme4.pdf

Jiang, Z. (2018). Using the linear mixed-effect model framework to estimate generalizability variance components in R: A lme4 package application. Methodology: European Journal of Research Methods for the Behavioral and Social Sciences, 14(3), 133–142. https://doi.org/10.1027/1614-2241/a000149

Jiang. Z., Raymond, M., Shi, D., & DiStefano, C. (2020). Using a linear mixed-effect model framework to estimate multivariate generalizability theory parameters in R. Behavior Research Methods, 52, 2383-2393. https://doi.org/10.3758/s13428-020-01399-z

We again thank the reviewer for taking the time to evaluate our manuscript and supplement and provide this useful feedback.